# Spatiotemporally distinct responses to mechanical forces shape the developing seed of *Arabidopsis*

Amélie Bauer [ID][1,2], Olivier Ali[1], Camille Bied[1], Sophie Bœuf [ID][1], Simone Bovio[1,3], Adrien Delattre [ID][1], Gwyneth Ingram[1], John F Golz [ID][2] & Benoit Landrein [ID][1✉]

## Abstract

**Organ morphogenesis depends on mechanical interactions between cells and tissues. These interactions generate forces that can be sensed by cells and affect key cellular processes. However, how mechanical forces, together with biochemical signals, contribute to the shaping of complex organs is still largely unclear. We address this question using the seed of *Arabidopsis as* a model system. We show that seeds first experience a phase of rapid anisotropic growth that is dependent on the response of cortical microtubule (CMT) to forces, which guide cellulose deposition according to shape-driven stresses in the outermost layer of the seed coat. However, at later stages of development, we show that seed growth is isotropic and depends on the properties of an inner layer of the seed coat that stiffens its walls in response to tension but has isotropic material properties. Finally, we show that the transition from anisotropic to isotropic growth is due to the dampening of cortical microtubule responses to shape-driven stresses. Altogether, our work supports a model in which spatiotemporally distinct mechanical responses control the shape of developing seeds in *Arabidopsis*.**

**Keywords** Morphogenesis; Seed Development; Microtubules; Mechanical Forces
**Subject Categories** Cell Adhesion, Polarity & Cytoskeleton; Plant Biology

## Introduction

Like all biological organisms, plants are physical structures whose growth depends on the generation of forces (Landrein and Ingram, 2019). At the cellular scale, plant growth relies on the balance between the hydrostatic pressure of the cell (i.e., turgor) and the mechanical properties of the surrounding cell wall. Pressure promotes growth by generating tension in plant cell walls. When tension exceeds the yielding threshold of the walls, it induces their plastic deformation, and thus irreversible cell expansion

(Boudaoud, 2010). Plant primary cell walls, which surround growing cells, are complex structures composed of stiff microfibrils of cellulose embedded in a gel-like matrix of pectins, of hemicelluloses, and a mixture of structural and regulatory proteins (Cosgrove, 2022). While the extensibility and the rigidity of the wall can be tuned by modifying the properties of any of these different components, its ability to stretch in a particular direction largely depends on the organization of the cellulose microfibrils, which are the main load-bearing structures of the wall. In this regard, cellulose microfibrils can be oriented in a particular direction to form crystalline arrays that resist stretching, forcing walls to expand perpendicularly to the main fibril axis (Cosgrove, 2022). Cellulose microfibrils are synthesized by multimeric transmembrane cellulose synthase complexes (CSCs) that move in the membrane following tracks composed of cortical microtubules (CMTs) (Paredez et al, 2006). By guiding the deposition of cellulose fibers, CMTs can determine the mechanical anisotropy of the wall and control the main orientation of cell elongation.

At the organ scale, plant growth patterns not only depend on the mechanical properties of each cell but also on mechanical interactions between cells and tissues. This is due to the presence of the wall, which glues cells together so that differences in mechanical properties between adjacent cells or adjacent layers generate mechanical conflicts that can induce the emergence of complex shapes when resolved (Rebocho et al, 2017; Coen et al, 2004). In several aerial organs such as meristems, stems, sepals or early-developing leaves, it is believed that growth is promoted by the pressure of inner tissues but restricted by the mechanical properties of one or several outer layers, which is often the epidermis (Kutschera and Niklas, 2007). This model of growth, defined as the "epidermal growth control" theory, is supported by both genetics and mechanical studies (Kelly-Bellow et al, 2023; Savaldi-Goldstein et al, 2007; Verger et al, 2018; Asaoka et al, 2021; Vaseva et al, 2018). This model also implies that cells experience specific patterns of mechanical forces that are not only dependent on their position within the organ but also on the overall shape of the organ (i.e., shape-driven stresses) (Hamant et al, 2008). Although the molecular mechanisms of mechanoperception are still largely unknown in the plant kingdom, accumulating evidence suggests that plant cells can sense mechanical signals, and use them as cues to assess shape changes occurring at the organ scale during

[1]Laboratoire Reproduction et Développement des Plantes, Université de Lyon, ENS de Lyon, UCB Lyon, CNRS, INRAE, INRIA, 69364 Lyon, Cedex 07, France. [2]School of Biosciences, University of Melbourne, Royal Parade, Parkville, VIC 3010, Australia. [3]Université Claude Bernard Lyon 1, CNRS UAR3444, Inserm US8, ENS de Lyon, SFR Biosciences, Lyon 69007, France. ✉E-mail: benoit.landrein@ens-lyon.fr

morphogenesis (Landrein and Ingram, 2019). Mechanical forces have notably been shown to affect CMT organization, and thus cellulose deposition (Hamant et al, 2008; Sampathkumar et al, 2014; Robinson and Kuhlemeier, 2018; Hejnowicz et al, 2000), so that shape-driven stresses can affect anisotropic growth in a variety of plant organs (Hervieux et al, 2016, 2017; Uyttewaal et al, 2012; Zhao et al, 2020). However, to what extent mechanical signals contribute to the control of plant organ shape remains unclear, partly because CMTs can also respond to other signals, such as light and hormones (Sambade et al, 2012; Vineyard et al, 2013), but also because organ shape is the product of a complex set of mechanical interactions between different cellular compartments, cells, and tissues, and does not only depend on the molecular mechanisms controlling cell growth anisotropy (Rebocho et al, 2017).

The seed of *Arabidopsis* is an excellent model system in which to study how mechanical interactions between layers contribute to plant organ growth. Seeds are composed of three genetically distinct compartments: the embryo, the endosperm and the surrounding seed coat (Ohto et al, 2007). As the growth of the seed precedes that of the embryo, it is thought to depend mainly on interactions between the endosperm and the seed coat. Accordingly, we recently provided evidence supporting that seed growth is promoted by the pressure of the endosperm but restricted by the mechanical properties of an internal layer of the seed coat, called the adaxial epidermis of the outer integument (ad-oi layer) (Creff et al, 2023, 2015; Beauzamy et al, 2016). We demonstrated that this layer stiffens its internal cell wall (called wall 3) in response to the tension induced by endosperm expansion and that this stiffening is associated with the accumulation of demethylesterified pectins in the wall. Using modeling and experiments, we showed that mechanosensitive stiffening of seed coat walls could underlie seed size control in *Arabidopsis* and explain the counter-intuitive effect of pressure on growth (Creff et al, 2023). However, although CMTs have been previously shown to respond to the application of mechanical forces in the seed coat outer integument (Creff et al, 2015), how this mechanism intersects with the mechanical feedback loop between CMTs and forces to determine seed shape, has not been investigated.

Here we show that seed growth patterns are not uniform and can be divided into two distinct phases: an early phase of rapid anisotropic growth followed by a phase of slower isotropic growth. We present evidence that CMT responses to forces in the abaxial epidermis of the outer integument of the seed coat (ab-oi, the outermost cell layer of the seed coat) drive the initial phase of anisotropic growth through the guided deposition of cellulose microfibrils in the outer wall of the seed. In contrast, we show that growth is restricted by the mechanical properties of wall 3 during the isotropic growth phase, that stiffens in response to forces, but should have isotropic material properties based on the organization of the CMTs in the adaxial epidermis of the outer integument (ad-oi). Finally, we show that the transition from anisotropic to isotropic growth does not depend on wall 3 stiffening but instead relies on a dampening of the response of CMTs to shape-driven stresses in the abaxial epidermis of the outer integument. Taken together, our work supports that the combined action and fine-tuning of two mechanical responses occurring in two different layers of the seed coat and at different times, control both seed growth rate and seed growth anisotropy, thus determining both seed size and shape.

# Results

## Seed growth is divided into an early phase of high anisotropic growth and a later phase of slow isotropic growth

To quantify the changes in size and shape that occur during seed development, we sampled fruits daily following anthesis, and extracted and optically cleared the developing seeds before measuring their size and shape on 2D images obtained by differential interference contrast (DIC) microscopy (Fig. 1A). As we previously reported (Creff et al, 2023), we found that seed growth occurred within the first week following fertilization. During this time, seed size increased following a characteristic S-shaped pattern: growth was higher at the beginning of the growth phase but slowly decreased until it terminated at around 7 days post anthesis (Fig. 1C). Seed shape resembles a flattened ovoid so that they tend to settle on one of their flatter sides when put between a slide and a coverslip, allowing accurate measurement of seed length and width in 2D. We thus studied the shape changes that occurred during seed growth by calculating the seed aspect ratio, which is defined as the ratio between the length of the seed and its width (see Methods and Fig. 1B). We observed two distinct growth behaviors. During the first 3 days following anthesis, seeds elongated along their future main axis so that the aspect ratio, which in ovules, was close to 0.8, increased rapidly, to reach a value close to 1.7. In contrast, between 3 and 7 days post anthesis, growth was much more isotropic as seeds expanded slightly more in width than in length (Figs. 1D and EV1A–C). We also looked at seed height by imaging seeds expressing an ubiquitous membrane marker (or a microtubule marker) by confocal microscopy, which allows 3D visualization. We observed that seeds are flattened ovoids at all stages considered (0, 2, and 5DPA) (Fig. EV1D). We focused on the 2DPA timepoint and measured the flatness of the seed by measuring seed width and thickness on median transverse sections of z-stacks perpendicular to the main seed axis and found that seed thickness was around 15% lower than seed width at this stage of development (Fig. EV1E). Taken together, these analyses show that seed growth involves two distinct phases: an initial phase of rapid anisotropic growth, where seeds mostly grow in length rather than in width and thickness, followed by a phase of slow and isotropic growth.

## Seed shape depends on the properties of the outer integument of the seed coat

We previously showed that seed size is the product of mechanical interactions between two seed compartments: the endosperm and the seed coat (Creff et al, 2023). In our mechanical model, the endosperm is predicted to generate an isotropic force via its hydrostatic pressure, whereas the cell walls of the seed coat could, if they have anisotropic mechanical properties, restrict growth in particular directions more than in others (Boudaoud, 2010; Creff et al, 2023). To test this hypothesis we examined the seed growth profile of mutant lines lacking functioning *KATANIN* (*KTN1*) or *SPIRAL2* (*SPR2*), two genes that normally encode for microtubule-associated proteins (MAPs) involved in the control of growth anisotropy (Burk et al, 2001; Shoji et al, 2004) and that affect, by severing microtubules for KTN1 or by acting on the severing

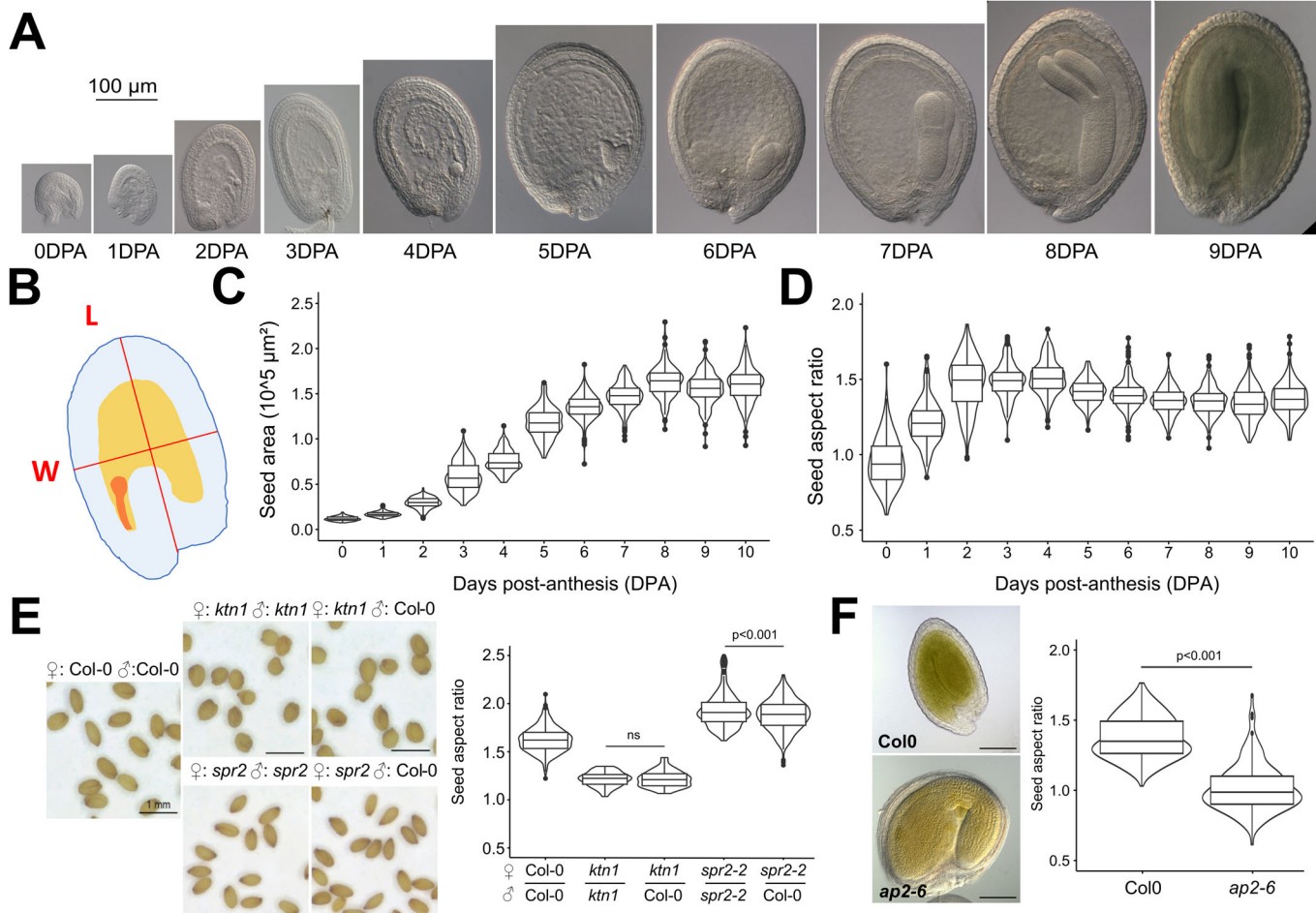

**Figure 1. Characterization of the anisotropic growth pattern of the seed and of its dependency on seed coat outer integument properties.**

(A). Representative WT seeds (Col-0 ecotype) from 0 to 9 days post anthesis (DPA), scale bar: 100 μm. (B) Seed aspect ratio is defined as the ratio between seed length (L, obtained by drawing a line from the pedicel of the seed to its tip that passes through the middle of the endosperm) and width (W, obtained by drawing a line perpendicular to the previous axis and passing through its center). The seed coat is highlighted in blue, the endosperm in orange and the embryo in red. (C, D) WT seed area (C) and aspect ratio (D) from 0 to 10DPA, $n = 180–209$ seeds per day, two independent experiments. In the boxplot representations, the midline represents the median of the data, while the lower and upper limits of the box represent the first and third quartile, respectively. The error bars represent the distance between the median and one and a half time the interquartile range. (E) Shape of mature seeds of *ktn1* and *spr2* mutants whose ovules were either self-pollinated or pollinated with WT pollen, scale bars: 1 mm, $n = 78–551$ seeds per genotype, two independent experiments. Data were compared using bilateral Student tests. In the boxplot representations, the midline represents the median of the data, while the lower and upper limits of the box represent the first and third quartile, respectively. The error bars represent the distance between the median and one and a half time the interquartile range. (F) Shape of Col-0 and *ap2* seeds (pollinated with WT pollen) at 10DPA, $n = 162–230$ seeds, two independent experiments. Data were compared using bilateral Student tests. In the boxplot representations, the midline represents the median of the data while the lower and upper limits of the box represent the first and third quartile, respectively. The error bars represent the distance between the median and one and a half time the interquartile range.

activity for SPR2 (Wightman et al, 2013), the organization of cortical microtubules (CMTs) in interphase cells, notably in response to mechanical forces (Uyttewaal et al, 2012; Eng et al, 2021; Hervieux et al, 2017, 2016; Lin et al, 2013; Sampathkumar et al, 2014; Zhao et al, 2020; Louveaux et al, 2016). We observed that self-pollinated plants produced seeds with similar shape defects (rounder seeds for *ktn1* mutants and more elongated seeds for *spr2* mutants) to those arising from mutant ovules pollinated with WT pollen. Given that this latter class of seeds have phenotypically wild-type endosperm and embryos (these tissues being heterozygous for the recessive *ktn1* or *spr2* mutations), but a phenotypically mutant seed coat owing to its maternal origin

(Fig. 1E), our data suggest that seed shape mainly depends on the behavior of the seed coat.

The seed coat is a maternal tissue that originates from differentiation of the ovule integuments and in *Arabidopsis*, is composed of four–five cell layers: the two outermost cell layers form the outer integument whereas the two–three innermost layers arise from the inner integument (Cucinotta et al, 2014). To test which layer of the seed coat determines seed shape, we characterized the shape of *apatela2* (*ap2*) mutant seeds, in which the outer integument cells fail to differentiate (Ohto et al, 2005; Jofuku et al, 2005). We observed that *ap2* seeds (produced by crossing *ap2* mutant plants with WT pollen to avoid zygotic effects)

were similarly round, showing that *AP2* affects seed growth anisotropy, and that seed shape depends on the properties of at least one layer of the outer integument (Fig. 1F). These results are in agreement with the epidermal growth control theory that proposes organ growth to be restricted by the properties of outer cell layers (Kutschera and Niklas, 2007).

## Cells in the outer integument of the seed coat also show two distinct growth patterns

As our results suggest that seed shape depends on the mechanical properties of the outer integument, we compared the growth pattern of the cells within the outer integument during the anisotropic growth phase (from 1 to 2DPA) and the isotropic growth phase (from 4 to 5DPA). To do this, we isolated seeds expressing an ubiquitous membrane marker from in vitro-grown fruits, by adapting a previously published protocol (Gooh et al, 2015), and imaged them by confocal microscopy at two different timepoints (0h and 24h). We then used the MorphoGraphX software and the level set method (LSM) to extract meshes of the outer surface of both layers of the outer integument, segment the

cells on these meshes and manually track them from one timepoint to the other (Barbier de Reuille et al, 2015; Kiss et al, 2017). The resulting growth maps of developing seeds enabled us to measure cell growth rate, growth anisotropy, and division rate in the two layers of the outer integument (Fig. 2A).

In agreement with our measurements at the organ scale (Fig. 1C), we observed that cells in both layers of the outer integument grew faster during the anisotropic growth phase than during the isotropic growth phase (Fig. 2A,B). Furthermore, we also found that growth was not homogenous in the outer integument, especially during the anisotropic growth phase, where cells within the top half of the seed (with regards to their main axis) expanded much more than the cells at the base and at the very tip of the seed (Fig. 2A,B), which is in line with observations for other plant organs such as leaves and sepals (Hervieux et al, 2016; Kuchen et al, 2012). A characteristic feature of the anisotropic growth phase was the preferential elongation of outer integument cells parallel to the main axis of the seed, whereas there was no preferential direction of growth during the isotropic growth phase, and during which cells acquired a soap bubble-like organization that is a defining mark of isotropic growth (Corson et al, 2009) (Figs. 2 and EV2A,B).

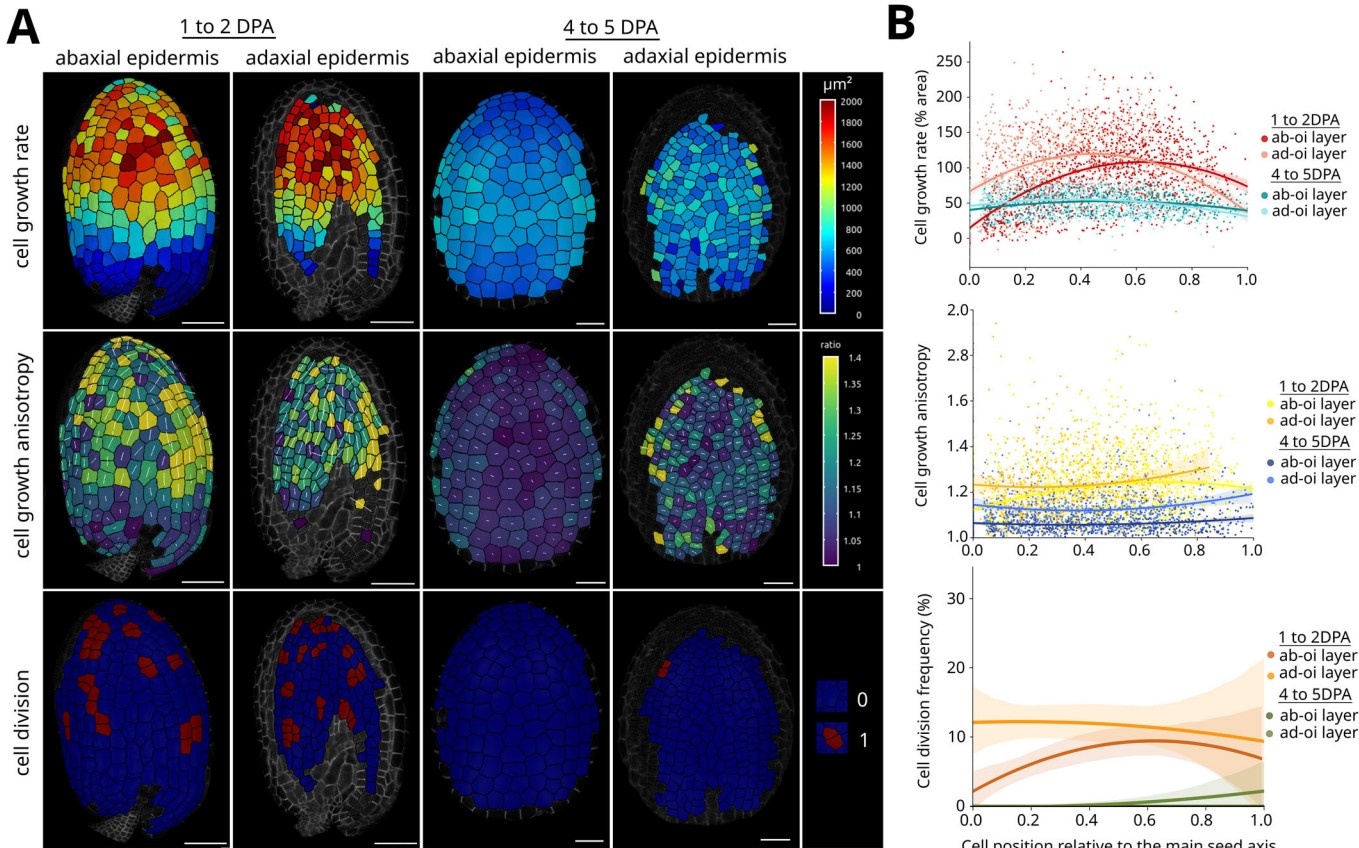

**Figure 2. Quantifying cell growth in the two layers of the seed coat outer integument.**

Representative maps (**A**) and quantifications (**B**) of cell growth rate, cell growth anisotropy and cell division in the abaxial (ab-oi) and adaxial (ad-oi) epidermis of the outer integument of the seed during the anisotropic growth phase (from 1 to 2DPA) and during the isotropic growth phase (from 4 to 5DPA). In the maps of growth anisotropy, the white lines overlaid on the heatmap represent the main axis of elongation of the cells. In the graphs in (**B**), the colored lines and dim bands show the mean and standard deviation as a function of the relative position along the main seed axis from the base to the tip. Scale bars: 20 µm, $n = 1015$–$1677$ cells from 10 to 11 seeds, two independent experiments.

Furthermore, we also observed that cells continued to divide during the anisotropic growth phase but not during isotropic growth phase so that during the latter, they greatly enlarged over time, especially in the abaxial epidermis (Figs. 2A,B and EV2A,B). We also quantified the changes in thickness of the cells in the ab-oi and ad-oi layers during development. As cells divide only anticlinally in both layers, the thickness of each layer only depends on growth. We thus studied the evolution of cell thickness by imaging different seeds expressing ubiquitous microtubule reporter and observed a 2.3–2.4-fold increase in the thickness of both layers from 0 to 5DPA (Fig. EV2C,D). This increase is relatively similar to the 2.7-fold increase in seed width observed from 0 to 5DPA, but is much lower than the 4-fold increase in seed length that is observed during this period (Fig. EV1A,B). In summary, our quantitative analysis demonstrates that the early anisotropic growth of the seed is associated with division and anisotorpic growth along the main seed axis of outer integument cells that are located at the top half of the seed, whereas the isotropic growth phase is associated with slow and isotropic growth of all cells throughout the outer integument.

## The anisotropic growth phase depends on CMT organization in the abaxial epidermis of the outer integument

Cortical microtubules are thought to determine the main axis of cell elongation by controlling the oriented deposition of cellulose microfibrils in the wall (Paredez et al, 2006). To confirm that this is the case in the *Arabidopsis* seed, we analyzed the growth pattern of seeds with defective microtubules and/or cell wall. This included the two mutants for genes involved in the control of microtubule organization that we previously described: *katanin* (*ktn1*) and *spiral2* (*spr2*), as well as mutants for two genes involved in anchoring CSCs to CMTs: *cellulose synthase interacting protein 1* (*csi1*) and *tetratricopeptide thioredoxin-like 1 and 3* (*ttl1 ttl3*), and a mutant lacking part of the primary cellulose synthase complex: *procuste1/cellulose synthase 6* (*prc1-1*) (Hervieux et al, 2016; Uyttewaal et al, 2012; Kesten et al, 2022; Fagard et al, 2000; Bringmann et al, 2012). We observed seed shape defects that correlated with the function of the associated genes for all of these mutants. Reducing CMT organization in *ktn1*, CMT guidance of the CSCs in *csi1* and *ttl1 ttl3*, or cellulose synthesis in *prc1-1* led to reduced growth anisotropy and the production of rounder seeds. In contrast, increasing CMT organization in *spr2* led to increased growth anisotropy and the production of more elongated seeds (Fig. EV3A,B). The analysis of the seed shape defects of mutants involved in the control of CMT organization and the guidance of CSCs supports a model in which CMTs control seed growth anisotropy through the guided deposition of cellulose microfibrils in the wall. However, which outer integument cell wall is load-bearing and determines the axis of elongation of the seed remains to be determined.

In developing pavement cells, CMTs in both longitudinal and transverse walls of the epidermal cells organize in specific patterns according to cell and tissue stresses to promote lobe formation (Belteton et al, 2021; Sampathkumar et al, 2014; Schneider et al, 2022). However, in organs in which epidermal growth control is at play, such as young leaves, sepals and meristems, it has been proposed that the CMTs facing the outermost longitudinal cell wall of the epidermis, which is believed to be load-bearing, organize

according to shape-driven stresses to control the anisotropic growth of these organs (Hamant et al, 2008; Hervieux et al, 2016; Sampathkumar et al, 2014; Zhao et al, 2020; Uyttewaal et al, 2012). In contrast, it is the CMTs that face the inner side of the epidermis that are robustly oriented perpendicularly to the main axis of elongation in hypocotyl (Crowell et al, 2011; Chan et al, 2011), suggesting that the inner wall of the epidermis is load-bearing and controls the elongation of this organ, a model which was further supported with computational simulations and experiments (Robinson and Kuhlemeier, 2018; Verger et al, 2018). To determine where seed growth anisotropy is controlled, we studied the organization of the CMTs in the two outer integument layers in the flatter faces of the seed as these are the regions that showed the highest growth rate in our maps. To do this, we imaged different microtubule reporters (ubiquitously expressed or specifically expressed in oi-ab or oi-ad), extracted the CMT signal near each longitudinal and transverse wall manually on confocal sections, or semi-automatically on 2.5D projections obtained using the MorphographX software, or on 2D projections obtained using the SurfCut plugin in ImageJ, and analyzed CMT organization using the FibrilTool plugin (Appendix Fig. S1) (Barbier de Reuille et al, 2015; Erguvan et al, 2019; Boudaoud et al, 2014).

We first compared the organization of the CMTs in the longitudinal walls facing the outer side of the seed in the abaxial epidermis (imaged using an ubiquitous microtubule reporter, *p35S::MAP65-1-RFP*) to the organization of the CMTs facing both inner and outer sides of the seed in the adaxial epidermis (imaged using a reporter we developped for this purpose, *pELA1::MAP65-1-mCitrine*) (Fig. 3A). We observed that the CMTs facing the outer face of the seed in the abaxial epidermis were highly organized and preferentially oriented perpendicularly to the main axis of the seed (Fig. 3A–C). In contrast, we observed that the CMTs facing either side of the seed in the adaxial epidermis were less organized and were not aligned in any particular direction (Fig. 3A–C). We also analyzed the organization of the CMTs facing the inner side of the seed in the abaxial epidermis using a line only expressing a microtubule reporter in this layer (*pPDF1::mCitrine-MBD*). Although this line shows CMTs that are less organized than those of the *p35S::MAP65-1-RFP* line (see below), we observed that the CMTs facing the inner side of the seed were also preferentially oriented perpendicularly to main seed axis, and that their orientation directly correlated to that of CMTs facing the outer face (Fig. 3D). We also looked at the organization of the CMTs in transverse walls of both abaxial and adaxial epidermis. We observed that the CMTs in transverse walls were preferentially oriented perpendicularly to the seed surface in both the abaxial and adaxial epidermis (Fig. 3E), where they may restrict the growth in thickness of these layers (Fig. EV2C,D). However, in the abaxial epidermis only, we observed differences in the number of CMT bundles in transverse walls depending on their orientation with respect to the main seed axis (Fig. 3E). Accordingly, we found that transverse walls that are perpendicular to the main seed axis shows an average CMT signal that is 1.4-fold stronger than that of transverse walls that are parallel to the main seed axis (Fig. EV3C). Taken together, these data suggest that the CMTs in the abaxial epidermis are organized like "hoops around a barrel" to restrict growth perpendicularly to the main seed axis through the guided deposition of cellulose fibers in the wall (Fig. EV3D). However, these data do not allow us to determine which longitudinal wall of the abaxial epidermis (outer or inner) is load-bearing.

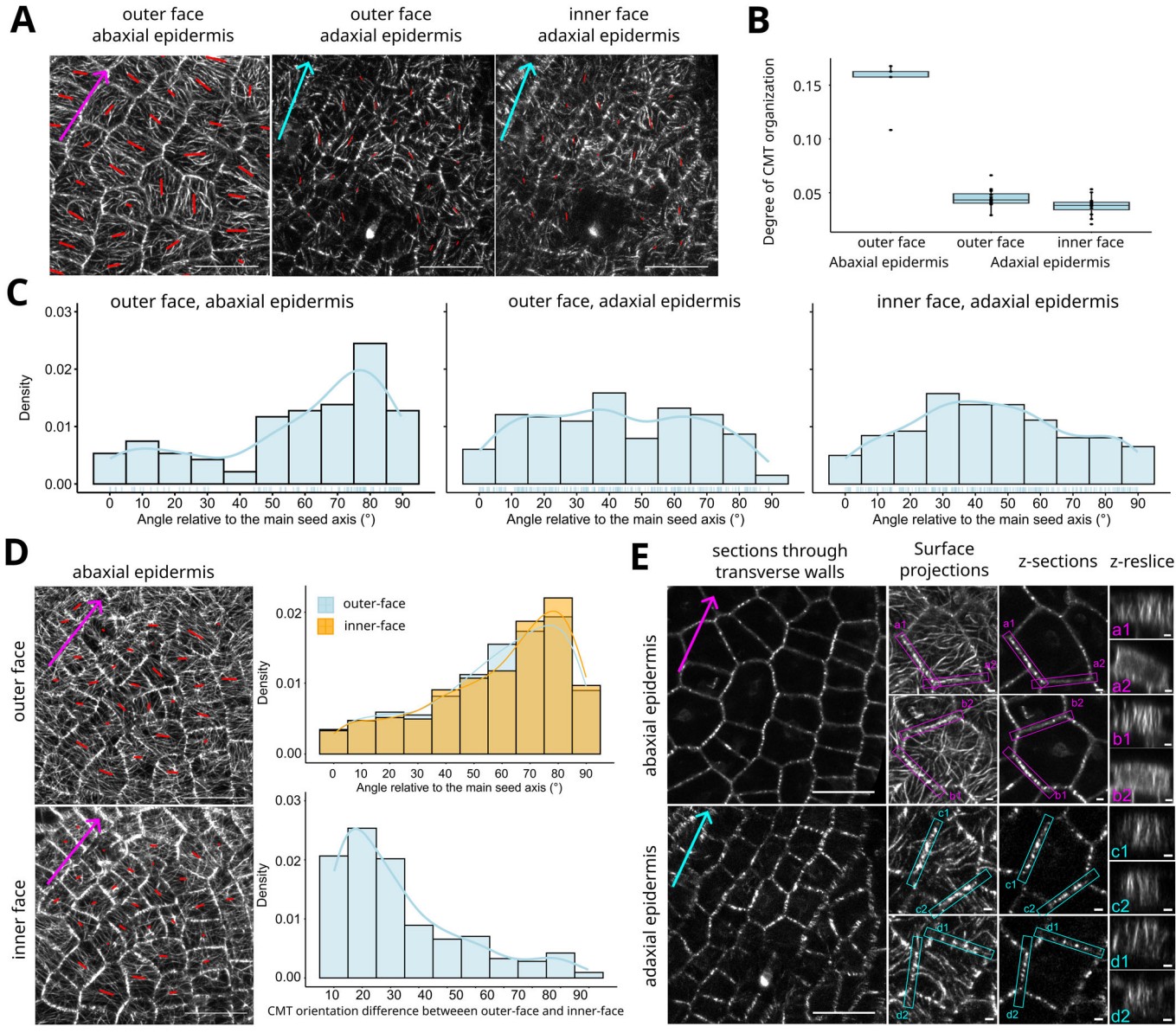

**Figure 3. Analysis of the organization of the CMTs in the abaxial and adaxial epidermis of the seed coat outer integument.**

(**A**) Visualization of the organization of the CMTs in the outer face of the abaxial outer integument epidermis (imaged using a *p35S::MAP65-1-RFP* reporter) and in both faces of the adaxial outer integument epidermis (imaged using a *pELA1::MAP65-1-mCitrine* reporter) during the anisotropic growth phase (2DPA). Each red bar shows the FibrilTool computation of the mean orientation of the CMTs per cell, and the colored arrows, the orientation of the main seed axis. Scale bars: 20 μm. (**B, C**) Degree of organization (**B**) and mean orientation relative to the main seed axis (**C**) of the CMTs in the outer face of the abaxial outer integument epidermis and in the two faces of the adaxial outer integument epidermis during the anisotropic growth phase (2 days post anthesis), $n = 94$–265 cells from 5 to 12 seeds, two independent experiments. In the boxplot representations, the midline represents the median of the data while the lower and upper limits of the box represent the first and third quartile, respectively. The error bars represent the distance between the median and one and a half time the interquartile range. (**D**) Correlation between the orientation of the CMTs in both faces of the abaxial outer integument epidermis (imaged using a *pPDF1::mCitrine-MBD* reporter) during the anisotropic growth phase (2DPA), $n = 491$ cells from 23 seeds, three independent experiments. In the pictures, each red bar shows the FibrilTool computation of the mean orientation of the CMTs per cell, and the colored arrow, the orientation of the mean seed axis. Scale bars: 20 μm. (**E**) Visualization of the organization of the CMTs in the transverse walls of the outer integument abaxial and adaxial epidermis (imaged using *p35S::MAP65-1-RFP* and the *pELA1::MAP65-1-mCitrine* respectively) from the seeds presented in (**A**). The colored arrows show the orientation of the mean seed axis, Scale bars, left side: 20 μm, right side (zoom on specific cells): 2 μm.

One of the evidence supporting that the outermost cell wall of the epidermis is load-bearing in the shoot apical meristem is its increased thickness compared to other walls (Kierzkowski et al, 2012). We thus studied the thickness of the walls of the outer integument at 3DPA by TEM. We observed that the longitudinal walls of the outer integument were much thicker than the transverse walls but also that wall 1, the outermost cell wall of the seed, was about 2-fold and 1.6-fold thicker than wall 2 and 3, respectively (Fig. EV3E). Although the relationship between wall thickness and rigidity is not straightforward, these data nonetheless

suggest that the outermost cell wall of the abaxial epidermis could, like in meristems, bear more load than other walls and thus be responsible for restricting growth during the anisotropic growth phase.

When imaging the abaxial epidermis, we observed striking differences in the CMT organization shown by different reporters. As mentioned above, CMTs were indeed less organized in the *pPDF1::mCitrine-MBD* reporter than they were in the *p35S::MAP65-1-RFP* reporter, while another CMT reporter, *p35S::TUA6-GFP*, showed CMTs whose degree of organization was intermediate compared to the two other reporters (Appendix Fig. S2A,B). This suggests that the presence of the reporter may affect the dynamics of the CMTs in the abaxial epidermis (and notably their bundling). To test whether this observed effect of the reporters on CMT organization affects the anisotropic growth of the seed, we compared the shape of mature seeds expressing the different reporters to that of seeds derived from control plants expressing a nuclear reporter in the abaxial epidermis of the outer integument (*pPDF1::CFP-N7* (Landrein et al, 2015)). This analysis revealed that the seeds expressing the *p35S::MAP65-1-RFP* reporter were 12% more elongated than control seeds while the seeds of the *pPDF1::mCitrine-MBD* reporter were 10% less elongated than control seeds (Appendix Fig. S2C), consistent with the differences in CMT organization between these lines. Given the influence of the CMT reporters on CMT organization and seed shape, our studies suggest that quantifications of CMT organization (bundling) should be interpreted with caution, as they may depend on the nature of the CMT reporter, while the measurements of CMT orientation seem to be reporter independent. The rounder shape of the seeds produced by the *pPDF1::mCitrine-MBD* reporter, which is only expressed in the outer integument abaxial epidermis, also provides further support for this layer being important for the control of seed growth anisotropy.

## CMT response to forces in the abaxial epidermis controls the elongation of the seed during the anisotropic growth phase

CMTs can repond to many different endogenous and environmental cues. However, accumulating evidence shows that mechanical forces play a key role in organizing CMTs during organ morphogenesis (Hamant et al, 2008; Hervieux et al, 2016; Sampathkumar et al, 2014; Zhao et al, 2020; Uyttewaal et al, 2012). In the seed, Creff and colleagues already showed that the CMTs of the abaxial epidermis become more organized when the fruit is compressed for 24 h (Creff et al, 2015), which we showed again in thus study (Fig. EV4A). To confirm this observation and study the response of CMTs to the application of mechanical constraints in a shorter timeframe, we performed ablations, which are known to generate circumferential stress and to reorient CMTs in the epidermis accordingly in other plant organs (Hamant et al, 2008; Sampathkumar et al, 2014). In the seed, we observed that the CMTs facing the outer face of the abaxial epidermis were able to reorient circumferentially around an ablation in about ~5 h (Fig. 4A). To further assess the response time of the CMTs to forces, we also used an indenter to perform a 20 µm indentation of the seed with a flat tip (1 mm) for very short periods of time (5–10 min, see "Methods"). We observed the disappearance of the CMT signal in the cells that were indented (at the top of the seed in

z-stacks) after indentation, followed by a circumferential reorientation of the CMTs in the surrounding cells after 5–6 h, which is the expected pattern for such indentation (Louveaux et al, 2016) (Fig. 4B; Appendix Fig. S3). Finally, in the controls of the two aforementioned experiments, seeds were grown in a liquid nutritive solution (see "Methods") which induces a fast and strong inflation of the seeds (Fig. EV4B). Interestingly, we observed that the increased growth induced by this treatment was correlated with an increased organization of the CMTs and of their tendancy to orient perpendicularly to the main seed axis in the outer face of the abaxial epidermis (Fig. EV4B). Taken together, these experiments provide support for mechanical forces affecting the orientation and degree of organization of the CMTs facing the outer face of the abaxial epidermis.

Having shown that CMTs facing the outer face of the abaxial epidermis can indeed respond to the application of mechanical forces, we investigated whether mechanical cues could expain the pattern of CMT organization observed in this layer. In developing seeds, the seed coat is thought to be put under tension by the expansion of the endosperm, that promotes growth via turgor pressure (Creff et al, 2015; Beauzamy et al, 2016; Creff et al, 2023). To further test this model, we performed endosperm pricking experiments (Movie EV1) to release endosperm pressure, which is known to strongly reduce the stiffness of early developing seeds (Beauzamy et al, 2016), and studied the resulting effect on seed coat structure (Fig. EV4C). In the abaxial epidermis, we observed a small but consistent decrease in cell length and width (around 2% and 2.5%, respectively) and a higher increase in cell thickness (around 10%) after endosperm pricking (Fig. EV4D). We also observed that the outermost cell walls of abaxial epidermal cells appeared more curved after pricking (Fig. EV4C). These data suggest that endosperm-derived turgor pressure increases tension in the longitudinal walls of the abaxial epidermis but relieves tension in the transverse walls.

In developing pavement cells, cell and tissue-driven stresses organize CMTs in both longitudinal and transverse walls in specific patterns to promote lobe formation (Belteton et al, 2021; Sampathkumar et al, 2014; Schneider et al, 2022). However, in tissues in which epidermal growth control is at play, shape-driven stresses are thought to predominate and to organize the CMT arrays accordingly (Hamant et al, 2008). To test if the shape of the seed, which appears to undergo epidermal growth control, prescribes a pattern of forces that is consistent with the organization of the CMTs observed in the outer integument abaxial epidermis, we developed a simple mechanical model where the seed is assimilated to an inflated shell whose shape is computed based on the characteristic length, width and thickness of seeds at 2DPA (Appendix Supplementary Methods). This model predicts that curvature and stress correlate both in terms of main directions and degree of anisotropy (Figs. 4C and EV4E). Moreover, because of the elongated shape of the seed, the model also predicts that in all areas of the seeds except the tip, both main stress and curvature directions are oriented perpendicularly to the main seed axis (Fig. 4C), which is the preferential orientation we observed for the CMTs in longitudinal walls of the abaxial epidermis (Fig. 3). To futher support this observation, we generated curvature maps of the surface of the seed so that we could correlate CMT orientation with local curvature anisotropy (Fig. 4D). We observed that the CMTs preferentially orient along the main axis of curvature of the seed at

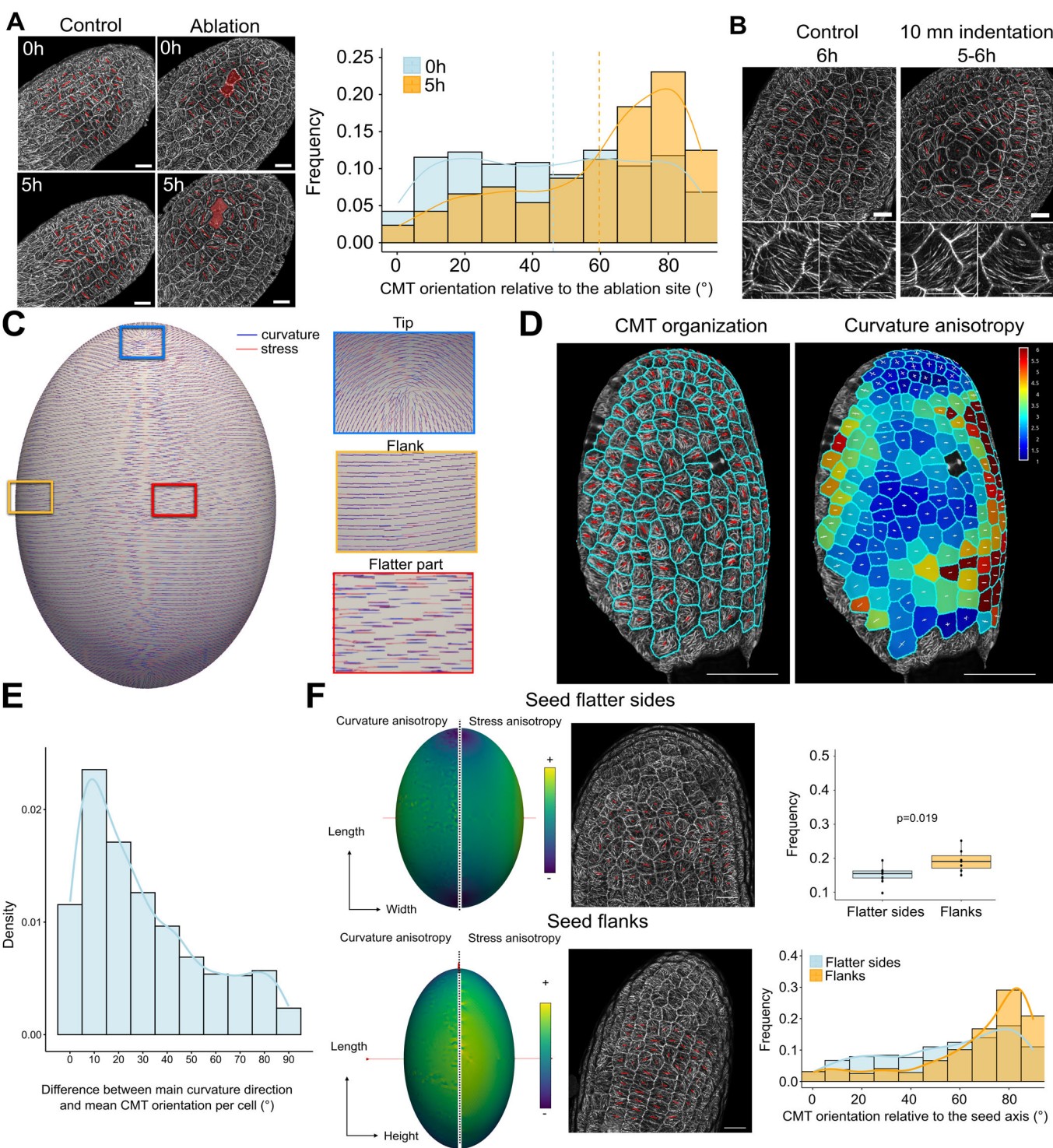

each cell location (Fig. 4E). Interestingly because the seed is flattened (Fig. EV1D,E), the model also predicted differences in the level of stress anisotropy between the flatter faces and the flanks of the seed. We thus compared CMT organization at these two locations and observed that the CMTs facing the outer face of the abaxial epidermis showed increased level of organization and were more prone to orient perpendicularly to the main seed axis in the

flanks, where more anisotropic stresses are experienced, than in the flatter faces of the seed (Fig. 4F). Finally, as the microtubule severing enzyme KTN has been shown to regulate the response to mechanical forces in several contexts (Uyttewaal et al, 2012; Eng et al, 2021; Hervieux et al, 2017, 2016; Sampathkumar et al, 2014), we also looked at the organization and response to mechanical forces of the CMT arrays in the *ktn1* mutant. We observed that the

**Figure 4.  CMTs responses to forces and organization according to shape-driven stresses in the outer face of the outer integument abaxial epidermis.**

(A) Effect of a cell ablation on the orientation of the CMTs (imaged using the *p35S::MAP65-1-RFP* reporter) facing the outer face of the seed in the abaxial outer integument epidermis, n = 425–751 cells from 11 to 16 seeds, three independent experiments. Scale bars, 20 μm. (B) Visualization of the reorganization of the CMTs facing the outer face of the outer integument abaxial epidermis 5–6 h after a 20 μm indentation of a 2DPA seed for 10 min (imaged using the *p35S::MAP65-1-RFP* reporter). Similar effects were observed on 8/10 indentations of 5–10 mn from two independent experiments (see "Methods" and Appendix Fig. S3). (C) Computer simulations showing the main stress orientations (red bars) and the main curvature orientations (blue bars) at the surface of an inflated shell whose elliptic shape has the same ratios of length, width and thickness as a typical 2DPA seed, see Appendix Supplementary Methods). (D, E) Correlation between the mean orientation of the CMTs in the outer face of abaxial epidermis (imaged using the *p35S::MAP65-1-RFP* reporter) and the anisotropy of the curvature of the seed during the anisotropic growth phase (2DPA), n = 2252 cells from 19 seeds, five independent experiments. In the CMT picture, the cell contours are highlighted in blue and the orientation of the red bars shows the mean orientation of the CMTs in each cell and its length, their degree of organization. In the heatmap of curvature anisotropy, the orientations of the two perpendicular white bars represent the axes of maximum and minimum curvature in each cell and their lengths the degree of curvature in each of these two directions. Scale bars: 50 μm. (F) Left: Differences in the degree of curvature anisotropy and of stress anisotropy between the flatter part and the flanks of seed in the computational simulation shown in (C). Right: Comparison of the mean degree of organization and of the orientation relative to the main seed axis of the CMTs facing the outer face of the outer integument abaxial epidermis in the flatter part, or in the flanks of the seed at 2DPA, n = 316–389 cells of 7–10 seeds, two independent experiments. Scale bars, 20 μm. Data were compared using bilateral Student tests. In the boxplot representations, the midline represents the median of the data while the lower and upper limits of the box represent the first and third quartile, respectively. The error bars represent the distance between the median and one and a half time the interquartile range.

rounder shape of the mutant we previously observed (Fig. EV3A) correlates with a decreased level of organization of the CMTs in the outer face of the abaxial epidermis, a lack of response to mechanical stimulations (following a 24 h compression of the fruit), and a reduction of the correlation between CMT orientation and main curvature direction (although this last observation should be interpreted with caution as the seed coat should experience less anisotropic stress in the *ktn1* mutant because of the rounder shape of its seeds) (Fig. EV4F). Taken together, these data all support the idea that CMTs in the abaxial epidermis organize according to shape-driven stresses to promote the elongation of the seed during the early anisotropic growth phase.

## The adaxial epidermis of the outer integument controls isotropic growth at late stages of seed development

Having provided strong evidence that CMT responses to forces in the abaxial epidermis of the outer integument promote growth anisotropy at early stages of seed development, we next investigated factors that control the isotropic growth of the seeds at later stages of development (from 3 to 7DPA). We recently showed that endosperm pressure directly promotes growth but indirectly inhibits it through the tension it generates in the seed coat, which promotes the stiffening of the walls of the inner face (wall 3) of the adaxial outer integument epidermis (Creff et al, 2023). Given that growth becomes completely isotropic around the time that wall 3 stiffens (at 3DPA, Fig. 1D), we considered whether the observed isotropic growth of the seed might arise from a transfer of load from the outer walls of the abaxial epidermis (wall 1 or 2) with their anisotropic material properties, to the walls of the inner face of the adaxial epidermis (wall 3) that have isotropic material properties.

To test this, we observed the organization of the CMTs in the inner and outer faces of the adaxial epidermis of seeds expressing the *pELA1::MAP65-1-mCitrine* reporter during the isotropic growth phase. In contrast to what we observed at 2DPA, we found that the CMTs facing the outer side of the seed in the adaxial epidermis were organized and preferentially oriented perpendicularly to the main seed axis, whereas the CMTs in transverse walls were still perpendicular to the seed surface and without any particular distribution with regard to the main seed axis (Fig. 5A,B). Moreover, we observed that the CMTs facing the inner side of the seed (i.e., facing wall 3) were much more disorganized compared to

those facing the outer side of the seed, and were preferentially oriented in parallel to the main axis of the seed (Fig. 5A,B). The lack of organization of the CMTs facing wall 3 is consistent with the hypothesis that wall 3, which is expected to be the main load-bearing wall during the isotropic growth phase (Creff et al, 2015; Creff, Ali et al, 2023), has isotropic material properties and controls isotropic growth at later stages of development.

The isotropic CMT organization we observed in the inner face of the adaxial epidermis could be associated with reduced mechanosensitivity of CMTs or be the consequence of reduced stress levels and/or stress anisotropy in this layer. To discriminate between these two possibilities, we examined the response of the CMTs to the application of mechanical forces in both sides of the adaxial epidermis. We observed that a 24 h compression of the fruit led to a strong increase in the organization of the CMTs facing the inner side of the seed but not of the CMTs facing the outer side, which were already organized (Appendix Fig. S4A,B). This demonstrates that the CMTs of the inner face of the adaxial outer integument epidermis still have the ability to respond to changes in stress pattern, and rather suggests that these CMTs are not organized either because stress is isotropic in this layer or because stress levels are too low to induce their organization.

To test if the transition from anisotropic to isotropic growth is a consequence of the stiffening of wall 3, we looked more closely at the correlation between seed growth rate and growth anisotropy, which was obtained by measuring seed size and aspect ratio. We reasoned that if these two parameters are correlated, then they might both depend on wall 3 stiffening, not only in the wild-type but also in *haiku2* (*iku2*), an endosperm-defective mutant in which higher endosperm pressure leads to more tension in the seed coat, inducing a precocious stiffening of wall 3 and early restriction of growth (Creff et al, 2023; Garcia et al, 2003). In the wild-type, we observed that both seed growth rate and seed growth anisotropy decreased over time but that seed growth anisotropy decreased earlier (peaking from 0 to 1DPA) than seed growth rate (peaking between 2 and 3DPA) (Fig. 5C; Appendix Fig. S4C). In the *iku2* mutant, seeds elongated more rapidly than WT seeds between 0 to 1DPA. This is expected as the seed coat of this mutant is under more tension than in WT seeds, which should enhance CMT organization during the anisotropic growth phase (Fig. 5C; Appendix Fig. S4C). However, after 1DPA, we observed that growth anisotropy was strongly decreased in *iku2*, reaching similar

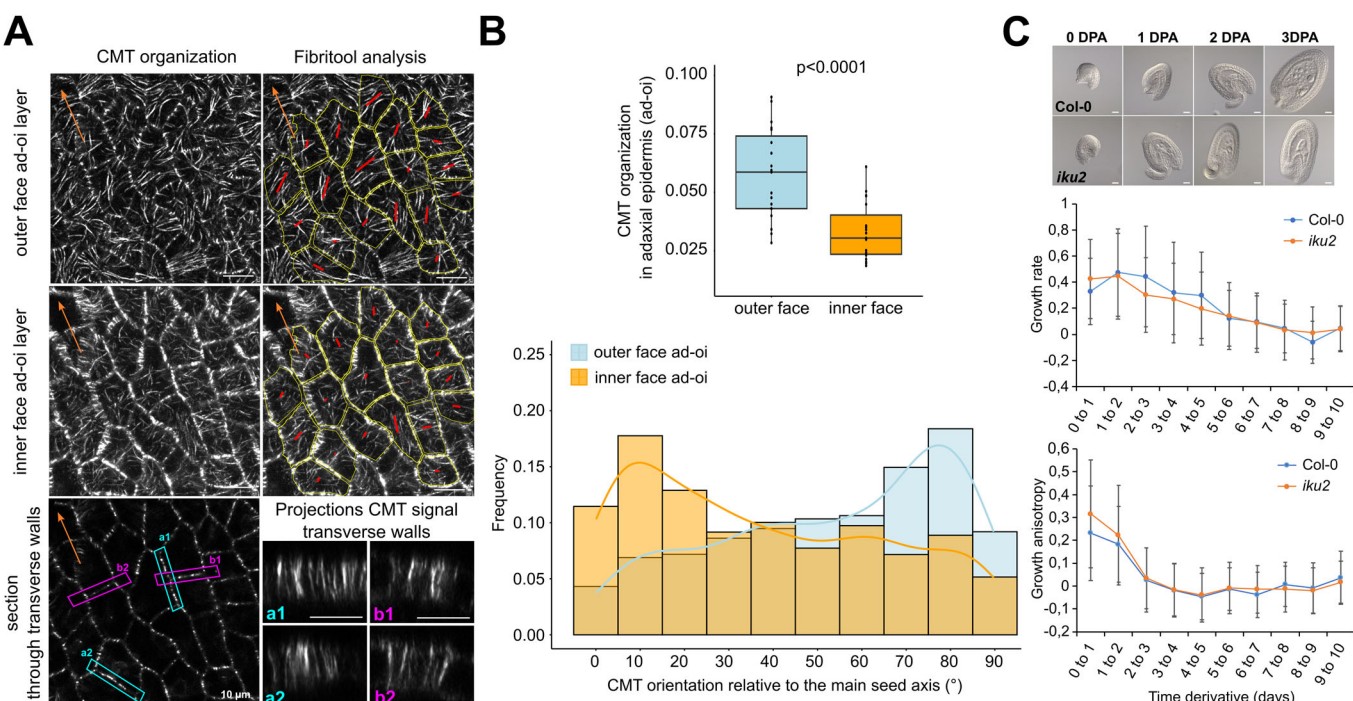

**Figure 5. Wall 3 stiffening controls isotropic growth at late stages of development but is not responsible for the transition from anisotropic to isotropic growth.**

(A, B) Representative images (A) and quantifications of the organization (B) of the CMTs (imaged using the *pELA1::MAP65-1-mCitrine* reporter) facing the inner and the outer face of the seed in the adaxial outer integument epidermis during the isotropic growth phase (5 days post anthesis (DPA). Scale bars: 20 µm, *n* = 395 cells from 19 seeds, three independent experiments. Data were compared using bilateral Student tests. The cell outlines are highlighted in yellow. Each red bar shows the FibrilTool computation of the mean orientation of the CMTs per cell, and the orange arrows, the orientation of the main seed axis. In the boxplot representations, the midline represents the median of the data, while the lower and upper limits of the box represent the first and third quartile, respectively. The error bars represent the distance between the median and one and a half time the interquartile range. (C) Left: Representative Col-0 and *iku2* mutant seeds at 0 to 3DPA, scale bars: 20 µm. Right: Average growth rate and growth anisotropy of developing WT and *iku2* mutant seeds from 0 to 10 days post anthesis (DPA) obtained from the measurements of seed size and aspect ratio presented in Fig. EVE5C, *n* = 207 to 313 seeds per day per genotype, three independent experiments. The error bars show the SD of the derivative (see "Methods").

levels to those observed of WT, and that this reduction in growth anisotropy also precedes the earlier restriction of growth observed in this mutant. These observations do not invalidate the idea that wall 3 is load-bearing and controls isotropic growth at later stages of development, but they show that the reduction of seed growth anisotropy observed between 1 and 3DPA is not a direct consequence of wall 3 stiffening.

## The transition from anisotropic to isotropic growth is associated with a disorganization of the CMTs in the outer face of the abaxial outer integument epidermis

If it is not a direct consequence of wall 3 stiffening, we reasoned that the transition from anisotropic to isotropic growth could be associated with a dampening of the CMT response to forces in the abaxial epidermis. To test this hypothesis, we compared the organization of the CMTs facing the outer face of the seed during both anisotropic and isotropic growth phase using the *p35S::MAP65-1-RFP* reporter. We observed that these CMTs were less organized and less prone to orient perpendicularly to the main seed axis at 5DPA than at 2DPA (Fig. 6A,B). With the same reporter, we also looked at the CMTs in the transverse walls of the abaxial epidermis and observed that they were still oriented

perpendicularly to the seed surface but that, contrary to what was observed at 2DPA, they were not differently distributed based on the orientation of the wall towards the main seed axis (Figs. 6A and EV3C,D). Similar observations regarding the orientation and degree of organization of the CMTs could be made with the *p35S::TUA6-GFP* reporter (Fig. EV5A–C). However, we could see a similar orientation but no decrease in CMT organization with the *pPDF1::mCitrine-MBD*, which is not surprising as the CMTs marked with this reporter are particularly disorganized during the anisotropic growth phase (hence the reduction of seed growth anisotropy observed in this mutant, Appendix Fig. S2). Nevertheless, we still looked at the CMTs facing the inner face of the seed using this reporter and observed that they had a similar orientation to, but a higher degree of organization than, those adjacent to the outer face (Fig. EV5D–F). These results indicate that the transition from anisotropic to isotropic growth is caused by a reduced organization of the CMTs in the outer face of the abaxial epidermis, although they can still, to some extent, orient according to shape-driven stresses (Fig. 6C). To further test this assumption, we imaged the CMTs of the outer face of the epidermis each day from 0 to 4DPA to characterize the dynamics of the disorganization. We observed that the CMTs were most organized at 0DPA but that their degree of organization linearly decreased over time (Fig. 6D),

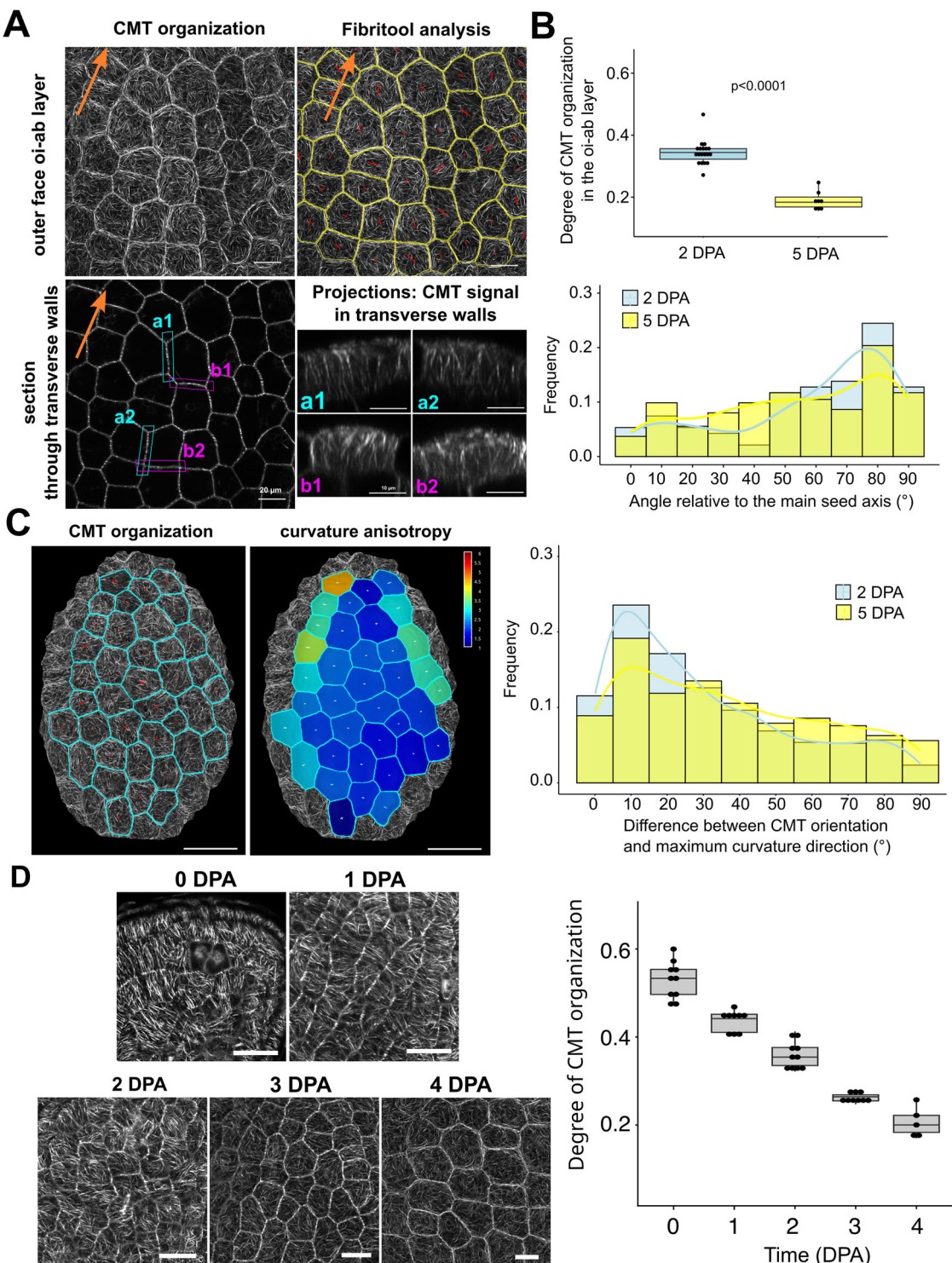

mirroring the observed decrease in growth anisotropy observed at the organ scale (Fig. 5E).

We also compared CMTs response to forces in the two faces of the abaxial epidermis using the *pPDF1::mCitrine-MBD* reporter to test if the reduced CMT organization in the outer face of the abaxial outer integument epidermis might be the consequence of reduced mechanosensitivity. Once again, we observed that the CMTs facing

the outer face of the seed, which were normally disorganized, responded strongly to seed compression, while the CMTs facing the inner side of the seed, which were already organized, did not show any increased organization in response to seed compression although they seem to form thicker bundles (Fig. EV5D–F). We also performed ablation studies to determine whether CMTs in the outer face of the abaxial epidermis respond to other types of

**Figure 6. The CMTs facing the outer face of the abaxial outer integument epidermis become disorganized during the transition to isotropic growth.**

(A) Organization of the CMTs facing the outer face and transverse walls of the abaxial epidermis at 5DPA (images using the *p35S::MAP65-1-RFP* reporter). Scale bars: 20 μm. The cell outlines are highlighted in yellow. Each red bar shows the FibrilTool computation of the mean orientation of the CMTs per cell, and the orange arrows, the orientation of the main seed axis. (B) Comparison between the degree of organization and the orientation relative to the main axis of the seed of the CMTs (imaged using the *p35S::MAP65-1-RFP* reporter) facing the outer side of the seed in the abaxial outer integument epidermis during the anisotropic growth phase (2DPA) and during the isotropic growth phase (5DPA). Scale bars: 20 μm, $n = 94–162$ cells from 5 to 10 seeds, two independent experiments. In the boxplot representations, the midline represents the median of the data while the lower and upper limits of the box represent the first and third quartile, respectively. The error bars represent the distance between the median and one and a half time the interquartile range. (C) Comparison between the orientation of the CMTs (imaged using the *p35S::MAP65-1-RFP* reporter) facing the outer side of the seed in the abaxial epidermis and curvature anisotropy of the seed during the anisotropic growth phase (2DPA, see Fig. 4B for representative pictures) and the isotropic growth phase (5DPA), scale bars: 50 μm, 2DPA: $n = 2252$ cells from 19 seeds, five independent experiments, 5DPA: $n = 303$ cells from 8 seeds, two independent experiments. In the CMT picture, each cell is highlighted in blue and the orientation of the red bars shows the mean orientation of the CMTs in each cell and their length, their degree of organization. In the heatmap of curvature anisotropy, the orientations of the two perpendicular white bars represent the axes of maximum and minimum curvature in each cell and their lengths, the degree of curvature in each of these two directions. (D) Evolution of the degree of organization of the CMTs facing the outer face of the abaxial epidermis in seeds from 0 to 4DPA, $n = 538–1450$ cells from 5 to 10 seeds, two independent experiments. Scale bars, 20 μm. In the boxplot representations, the midline represents the median of the data, while the lower and upper limits of the box represent the first and third quartiles, respectively. The error bars represent the distance between the median and one and a half time the interquartile range.

mechanical perturbations. Like at 2DPA, we observed a circumferential reorientation of the CMTs around the ablation site after 6 h (Fig. EV5G). These data indicate that the transition from anisotropic to isotropic growth is not associated with a reduced mechanosensitivity of the CMTs in the outer face of the abaxial outer integument epidermis but could be due to a reduction in stress levels or/and anisotropy.

## Discussion

Seed growth depends on mechanical interactions between the zygotic endosperm and the surrounding maternally-derived seed coat. We previously showed that endosperm pressure both promotes and restricts seed growth through the tension it generates in the seed coat, which induces wall stiffening in the adaxial epidermis of the outer integument (Creff et al, 2023). Results from the current study extend this model by showing that endosperm pressure controls seed shape by triggering sequential and different responses in neighboring cell layers of the outer integument. At early stages of development, CMTs respond to shape-driven stresses in the abaxial epidermis by preferentially orienting perpendicularly to main seed axis, resulting in the elongation of the seed. At later stages of development, endosperm pressure induces stiffening of the wall facing the inner side of the adaxial epidermis (wall 3). This wall subsequently becomes load-bearing and, based on the organization of its CMTs, controls growth isotropically. Finally, our results show that the transition from anisotropic to isotropic growth does not depend directly on the mechanosensitive stiffening of wall 3, but likely involves a dampening of the response of the CMTs facing the outer side of the seed in the abaxial epidermis to shape-driven stresses, eventhough these CMTs remain mechanosensitive (Fig. 6E).

Although our results are consistent with the early anisotropic growth of the seed being driven by CMT responses to forces in the abaxial epidermis of the outer integument, previous work has suggested that the feedback between CMTs and forces amplifies growth anisotropy but is not necessarily its initial trigger. For instance, it has been proposed that the elongation of developing hypocotyls is promoted by CMT response to forces and subsequent cellulose deposition according to tissue stresses, but is initiated as a result of an asymmetric distribution of methylesterified and demethylesterified pectins between transverse and longitudinal walls in the epidermis (Peaucelle et al, 2015). Although this still needs further experimental proofs, it has also been proposed that differences in mechanical properties between contiguous transverse walls in pavement cells could be associated with local pectin accumulation, which may trigger the initiation of the lobes through buckling (Majda et al, 2017), that would then be amplified through CMT responses to forces (Sampathkumar et al, 2014). In lateral roots, elongation depends on CMT guidance of cellulose deposition but also on an independent membrane trafficking pathway that occurs at cell edges and requires the activity of the small GTPase Rab-A5c (Kirchhelle et al, 2019). Finally, in leaves and sepals, CMT responses to force are necessary for the maintenance of organ flatness, but flatness also requires an initial degree of organ asymmetry generated by the influence of margin-specific genes on early primordium growth (Zhao et al, 2020). The ovules of *Arabidopsis* have the shape of flattened ovoids (Vijayan et al, 2021), which intuitively, should elongate in a direction opposite to that observed in fertilized seeds. Our results show that early-developing seeds exhibit an apicobasal growth gradient. Whether this initial asymmetry in growth, acting on the ovoid shape of the ovule, is sufficient to generate a pattern of mechanical forces that induces the elongation of the seed along its main axis, or requires the activity of specific growth regulators such as *APETALA2*, remains to be demonstrated.

The ability of CMTs to respond to forces and to align along the main axis of tensile stress has been observed in a variety of plant organs, such as leaves, hypocotyls, sepals and meristems, and even in confined protoplasts (Hamant et al, 2008; Zhao et al, 2020; Sampathkumar et al, 2014; Colin et al, 2020; Robinson and Kuhlemeier, 2018). It has thus been proposed that CMTs could, by themselves, act as tension sensors (Hamant et al, 2019). However, this view might be oversimplistic as we know that CMTs can also respond to other signals, such as light and hormones (Sambade et al, 2012; Vineyard et al, 2013), and that their organization does not always correlate with predicted stress patterns. This is notably the case in hypocotyls, where the CMTs of the outer face of the epidermis are not aligned along the axis of maximum tension (i.e., perpendicularly to the axis of the hypocotyl) during the entirety of the growth phase (Chan et al, 2011; Crowell et al, 2011). Our comparative analyses of CMT organization and response to forces in the outer integument of the seed coat also show that the CMTs

facing the different faces of outer integument cells do not necessarily organize according to predicted shape-driven stresses, even if they can still respond to the application of mechanical forces.

There may be several explanations for these intriguing results. The stresses experienced when seeds are compressed, or when an ablation is performed, could be different, both qualitatively and quantitatively, to those perceived when seed coat walls are put under tension by the pressure of the expanding endosperm. This is consistent with recent work showing that CMTs respond differently to tensile forces and to compressive forces during hypocotyl-stretching experiments (Robinson and Kuhlemeier, 2018). This is also consistent with work in protoplasts showing that CMTs preferentially orient along the main stress direction when pressure is high, but to cell geometry when pressure is low (Colin et al, 2020). We could hypothesize that the stiffening of wall 3, which is an internal wall, may isolate the outer integument from the tension induced in the seed coat by endosperm expansion (Creff et al, 2015). This would explain why the CMTs of the outer side of the abaxial epidermis (facing wall 1) are organized during the anisotropic growth phase, when this wall is load-bearing, but not during the isotropic growth phase, when wall 3 becomes load-bearing. However, this does not explain why seed growth anisotropy decreases before wall 3 starts to stiffen, or why the CMTs that face wall 2, in both abaxial and adaxial epidermis, remain organized at later stages of development, while the CMTs that face the load-bearing wall 3, are not particularly organized. Another possibility could be that tension is reduced in specific walls of the outer integument as a result of changes in their thickness and/or stiffness caused by their differentiation. This possibility is supported by our immunolabelling studies, showing that antibodies targeting pectins with different degrees of methylesterification do not mark outer integument walls uniformly, and that the labeling also depends on the developmental stage considered (Creff et al, 2023). This view is also consistent with studies showing that CMT organization and response to stresses can be enhanced by altering cell wall composition, for instance by inhibiting cellulose deposition in meristems (Heisler et al, 2010) or by altering pectin methylesterification in pavement cells (Tang et al, 2022). Nevertheless, our study supports the idea that CMT responses to stresses are key determinants of seed shape and that variations in these responses, in different outer integument layers, or at different stages of development, underlie the changes in growth anisotropy that are observed during seed development.

# Methods

## Plant material and growth conditions

The *spr2-2* (CS6549 (Shoji et al, 2004)), *ktn1* (SAIL_343_D12 (Chen et al, 2014)), *ap2-6* (CS6241 (Wakem, 2003)), *iku2-2* (Garcia et al, 2005), *csi1-3* (SALK_138584 (Lei et al, 2014)), *prc1-1* (CS297 (Fagard et al)) and *ttl1 ttl3* (Salk_063943 and Sail_193_B05 (Lakhssassi et al, 2012)) mutants were described previously. The *p35S::Lti6b-GFP* (Cutler et al, 2000), *p35S::MAP65-1-RFP* (Creff et al, 2015), *p35S::TUA6-GFP* (Ueda et al, 1999), *pPDF1-MBD-mCitrine* (Malivert et al, 2021) and *pPDF1::CFP-N7* (Landrein et al, 2015) reporters were also described previously. The *pELA1::MAP65-1-mCitrine* reporter

was developed for this study (see below). Seeds were sterilized in a solution of 70% ethanol and 0,05% Triton x100 (Sigma) for 15 min, rinsed three times in Ethanol 95%, and dried on Whatman paper for 30 min to 1 h under the hood. Seeds were then sowed on plates containing 1× Murashige and Skoog (MS) medium (pH 5.7) and kept for two days in the dark at 4 °C for stratification. They were then placed in a growth cabinet (Sanyo) under short-day condition (8 h light, 21 °C on daytime and 18 °C on nighttime). After 2 weeks, germinated seedlings were transferred into individual pots of soil (Argile 10 (Favorit)), placed in a short-day room (8 h light, 21 °C on daytime and 19 °C on nighttime) for a week before being transferred in a long-day room for the rest of their lifecycle (16 h light, 21 °C on daytime and 19 °C on nighttime). Note that the seeds used for the ablation experiment (Fig. EV5B) and the imaging of microtubules using the *pPDF1-MBD-mCitrine* line (Figs. 5 and EV3; Appendix Fig. S4) were grown differently. Following sowing on plates and stratification, they were placed in a long-day growth cabinet (16 h light, 21 °C at day and 19 °C at night) and transferred directly in a long-day room (16 h light, 21 °C at day and 19 °C at night) after germination.

## Generation of the *pELA1::MAP65-1-mCitrine* line

The *pELA1* promoter was amplified and cloned into the pENTRY-R4-L1 as described in (Creff et al, 2015). A triple Gateway reaction (Life Technologies) was then performed using the *pELA1::pENTR-R4-L1* plasmid, a *MAP65-1-pENTR-L1-L2* plasmid and a *mCitrine-pENTR-R2-L3* plasmid as entry vectors and the *pS7m34GW* plasmid as destination vector to generate a *pELA1::MAP65-1-mCitrine-pS7m34GW* construct. Plants were transformed by flower-dipping using *Agrobacterium*. Transformants were selected based on their Sulfadiazine resistance.

## Measurements of mature seed shape

Plants were grown under the conditions described in the previous section. Upon harvest, mature seeds were imaged using a SMZ18 Stereomicroscope (Nikon) mounted with a C11440 digital camera (Hamamatsu). The resulting pictures were analyzed using a dedicated macro on ImageJ. This macro uses a Huang thresholding to binarize the images and a distance-transformed watershed algorithm to segment the seed contours and separate adjacent seeds. The seeds that were incorrectly segmented were manually removed from the analysis. For each seed, the aspect ratio was defined as the ratio between the major and the minor axis of the seed, which was automatically obtained using the "Fit Ellipse" option in ImageJ.

## Measurements of the size and shape of developing seeds

For all experiments but the reciprocal crosses and the one involving *ap2-6*, the seeds were staged every day for up to 10 days by marking the opening of the flowers on the main stem with colored cotton threads. For the comparative analysis of Col-0 and *ap2-6* seed shape at 10DPA and the reciprocal crosses, both WT and mutant flowers were emasculated before their opening and manually pollinated with Col-0 pollen to ensure that the seed growth defects were of maternal origin. The siliques were then harvested, put on double-sided tape and opened with a needle so that their replum could be removed with forceps. The seeds on their replum were then put on

a slide with a drop of clearing solution (1 vol glycerol/7 vol chloral hydrate liquid solution, VWR Chemicals). The seeds were detached from the replum and the slide was covered with a coverslip and kept in the dark for at least 24 h in a cold room (4 °C). The seeds were imaged with an Axioimager 2 (Zeiss) equipped with 10× and 20× DIC dry objectives and mounted with a Axiocam 705 color camera (Zeiss).

The measurements of seed size and shape were done using a semi-automatic Fiji macro. To measure seed area, the seed contour was extracted automatically using a Huang Dark auto-thresholding and a distance-transformed watershed algorithm, or manually using the Polygon selection tool of ImageJ (for older seeds when the automatic segmentation was failing and for young seeds when the pedicel was still attached to the seed and could not be removed from the segmentation). The aspect ratio of the seed was defined as the ratio between the length and the width of the seed, which were measured manually using the straight-line selection tool of ImageJ. The length of the seed was obtained by drawing a line from the pedicel of the seed to its tip that passes through the middle of the endosperm sac and its width was obtained by drawing a line perpendicular to the previous axis and passing through its center.

### Preparation of the samples for live-imaging

For all experiments involving confocal microscopy, except for the analysis of cell growth that required a specific protocol to keep seeds growing over long period of time (24 h), seeds expressing membrane or microtubule reporters were grown as described in previous sections and the opening of the flowers of the main stems was marked with a cotton thread. On the day of imaging, the siliques were harvested and put on a slide covered with double-sided tape, and opened with a needle. The seeds were then gently taken by their replum with tweezer and put on double-sided tape in a small imaging plate before being covered with liquid culture medium (1/2 MS (Sigma), 1% Sucrose (Sigma), 1% Gamborg Vitamins (Sigma)). The seeds were imaged straight away or 5 h after preparation (for the endosperm pricking experiment).

### Measurements of microtubule organization in the seed outer integument layers

The *p35S::MAP65-1-RFP*, *pPDF1-MBD-mCitrine* and *p35S::TUA6-GFP* reporters were used to image the CMTs of the outer face of the abaxial outer integument epidermis. The *pPDF1-MBD-mCitrine* was also used to compare CMT organization in both faces of the abaxial epidermis, being only expressed in this layer. The *pELA1::MAP65-1-mCitrine* reporter was used to compare CMT organization in both faces of the adaxial epidermis, being only expressed in this layer. All acquisitions were performed using a Zeiss 980 Airyscan microscope with a water-dipping plan apochromat 20× objective (NA = 1), except for the seed compressions at 2DPA where a SP8 confocal microscope (Leica Microsystems) with a 25× long-distance water-dipping objective (NA = 0.95) was used, and the cell ablations and the imaging of the CMTs facing of the abaxial epidermis using the *pPDF1::MBD-mCitrine* line, where a Nikon C2 with a 60× water-dipping objective was used (NA = 1). The GFP was excited with a LED laser emitting at a wavelength of 488 nm (Leica Microsystems) and the signal between 495 nm and 555 nm was collected. The RFP was

excited with a LED laser emitting at a wavelength of 552 nm and the signal was collected at 580–650 nm. The mCitrine was excited with a LED laser emitting at a wavelength of 514 nm, and the signal was collected at 520–560 nm.

The mean orientation and organization of the CMTs per cell were measured with the FibrilTool macro in ImageJ (Boudaoud et al, 2014), either manually on confocal sections or semi-automatically using 2D projections of the CMT signal in the outer face or inner face of the cell using the SurfCut plugin as described in (Erguvan et al, 2019) (Appendix Fig. S1), except for the measurements of CMT organization following seed compression at 2DPA (Fig. 4) and for the measurements of the correlation between CMT orientation and local seed curvature, that were done on 2.5D meshes using the FibrilTool macro integrated in the MorphographX software. Although we checked that the degree of organization of the CMTs calculated by FibrilTool does not appear to depend on the type of microscope used for imaging (Appendix Fig. S1), the type of images that are used (z-sections, 2D projections obtained using CutSurf, or 2.5D projections obtained using MorphographX) seems to affect this parameter. The same type of image analysis was thus done each time degrees of organization were compared. Also note that the brightness and contrast of the images were enhanced in the figures for a better visualization of CMT organization.

### Analysis of the correlation between mean CMT orientation and local curvature direction

To generate the heatmaps and frequency plots showing the correlation between the orientation of the CMTs in the outer face of the abaxial epidermis and the local direction of maximum curvature at each cell position, maps of curvature anisotropy, projected on the cell-segmented 2.5D meshes, were generated with MorphographX. Given that the cells in the abaxial epidermis enlarge from 2 to 5 days post anthesis (DPA), a neighboring of 30 μm was used to generate the curvature maps at 2DPA while a neighboring of 60 μm was used to generate the curvature maps at 5DPA. This neighboring roughly corresponds, in both cases, to the distance between the center of the considered cell and the furthest edge of its direct neighbors. The angle θ representing the difference between CMT orientation (defined as a vector $(x_{Mt}, y_{Mt}, z_{Mt})$) and maximum curvature direction (defined as a vector $(x_{Mc}, y_{Mc}, z_{Mc})$) was calculated using the following formula:

$$\theta = \cos^{-1}(|x_{Mt} \cdot x_{Mc} + y_{Mt} \cdot y_{Mc} + z_{Mt} \cdot z_{Mc}|)$$

Note that both vectors are tangent to the surface of the mesh at the center of each cell considered.

### Analysis of cell growth in 2.5D using the MorphographX software

Plants expressing the ubiquitous membrane marker *p35S::LTi6b-GFP* were grown as described in previous sections and the opening of the flowers of the main stems was marked with a cotton thread. At either 1 or 4 days post anthesis, the siliques were harvested and put on a slide covered with double-sided tape, and opened with a needle (under the hood, using sterilized material). The replum with its seeds attached was then extracted with tweezers and put on

plates containing 2% Nitsch medium (Duchefa), 5% Sucrose (Sigma), 0.05% MES (Sigma), 1% agarose (Sigma), 1% Gamborg Vitamins (Sigma) and 0,1% Plant Preservative Mixture (PPM, Plant Cell Technology). Note that the medium (pH 5.8) needs to be autoclaved using a soft cycle (110 °C for 10 min) and that the Gamborg vitamins and the PPM need to be added after autoclaving. Once put on the culture medium, the seeds, still attached to their replum, were covered with a drop of 0.5% Low Melting Agarose (Sigma) to prevent movement during imaging. The plates were then closed with plaster and placed in a long-day growth cabinet (Panasonic, 16 h light, 20 °C) until imaging. To image the seeds, the plates were opened and covered with sterile water. Z-stacks (1 µm thickness) were collected using a SP8 confocal microscope (Leica Microsystems) with a 25x long-distance water-dipping objective (NA = 0.95). The GFP was excited with a LED laser emitting at a wavelength of 488 nm (Leica Microsystems), and the signal between 495 nm and 555 nm was collected. After imaging of the first timepoint, the water was removed, the plates were closed, sealed again, and put back in the growth cabinet for 24 h.

The resulting 3D stacks of developing seeds were analyzed using the MorphographX software (Barbier de Reuille et al, 2015) following the protocol described in https://www.mpipz.mpg.de/4085950/MGXUserManual.pdf. The level set method (Kiss et al, 2017) was used to generate a 2.5D mesh of the surface of the seed on which the signal of the transverse walls of the abaxial epidermis was projected. To obtain the 2.5D surface of the adaxial epidermis, the surface obtained for the abaxial epidermis by the level set method was translated in z of -10 µm and the level set method (option "evolve") was used again for several cycles to detect the longitudinal wall separating the abaxial from the adaxial epidermis (wall 2). A 2.5D mesh of this new surface was then generated, on which the signal of the transverse walls of the adaxial epidermis was projected. Note that this method generates a 2.5D mesh of the surface of the adaxial epidermis that is of a lower quality than the one of the abaxial epidermis. In both layers, the cells were then manually seeded and their contour was segmented using the watershed algorithm of the software. The parent-to-offspring association (lineage) of the cells of two successive timepoints was done manually according to the software guide. Heatmaps of cell size, aspect ratio, growth rate and growth anisotropy were then generated according to the software guide. The data were exported as csv files and analyzed on R (https://www.r-project.org/). The graphs displaying growth rate and anisotropy as a function of the position of the cell along the main axis of the seed were generated as described in: (Strauss et al, 2022).

### Endosperm pricking experiments and analysis

The endosperm pricking experiments were performed on 2DPA seeds expressing the *LTi6b-GFP* reporter that were put on double-sided tape in a plate and covered with liquid culture medium for 5 h before the experiments (to avoid the influence of the liquid medium on seed growth observed in Fig. EV4B), before being imaged using a Leica SP8 confocal microscope. Each seed was imaged a first time (before pricking), then a needle was used to prick the endosperm, and the seed was directly imaged a second time (after pricking). The experiment was conducted on each seed separately to reduce the time between the acquisition of the two stacks, which was only of around 5 min. The seeds were kept in liquid medium all experiment long (even during pricking) to avoid dehydration.

The resulting confocal z-stacks were analyzed using the MorphographX software (Barbier de Reuille et al, 2015) and according to guideline book for 3D segmentation and tracking (https://www.mpipz.mpg.de/4085950/MGXUserManual.pdf). First, the z-stacks were blurred (1 µm in each direction) and the cells were segmented using the ITK 3D segmentation tool. A 3D cell mesh (size 1 µm) was then generated and the cells of the abaxial epidermis of the growing region of the flatter part of the seed were manually selected. The tracking of the cells was then done on the 3D cell meshes by manually selecting, for each cell before pricking, the corresponding cell after pricking. Bezier coordinates were then used as described by (Strauss et al, 2022) and in the MorphographX guideline book so that cell segmented in 3D could have their length (i.e., longitudinal length), width (i.e., circumferential length), and thickness (i.e., radial length) defined in regards to the seed surface and main axis of elongation. Note that this was done following the same pipeline as the one described for the root, a choice that was motivated by the fact that the seed has a flattened but elongated shape at 2DPA. The changes in length, width, and thickness of the cells before and after pricking were then computed as percentages of increase/decrease between timepoints. As few cells in each seed had unrealistic values of increase/decrease in length, width or/and thickness because of segmentation errors, the median of all the cells per seed was considered in the quantifications instead of the mean.

### Application of mechanical constraints to developing seeds

For the compression, the entire silique was compressed (at 1 or at 4DPA) using a microvice as described in (Creff et al, 2015) and kept for 24 h in a growth chamber before preparation of the seeds and subsequent imaging of the CMTs as described in the previous sections. Optical sections of the 3D stacks were generated to check that the seeds were efficiently compressed.

For the ablation, the seeds were prepared as described in the live-imaging section. The ablation was done manually using a sharp needle under a stereomicroscope as described in: (Uyttewaal et al, 2012). Seeds were kept in a closed box between two timepoints. The measurements of the orientation of the CMTs in each cell relative to the position of the ablation was done in 2D on ImageJ as described in (Malivert et al, 2021).

For the indentation, the seeds were prepared as described in the live-imaging section. They were first imaged using a Zeiss 980 Airyscan. They were then indented with a Hysitron TI950 triboindenter (Bruker) equipped with a flat ended sapphire tip, 1 mm in diameter (TI-0068). The sample displacement was obtained using a 500 µm piezoelectric stage (NanoCZ500, Mad City Labs Inc). During the measurement, the seed was compressed twice: in each of these cycles, the sample was compressed up to 20 µm in 10 s, the tip position was maintained constant for 10 s and then the tip was lifted back by 20 µm in 10 s. After a pause of 10 s at the end of the second unloading phase, the sample was compressed again by 20 µm and the compression was maintained for either 5 or 10 min. After indentation, the seeds were kept in culture medium before being imaged again by confocal microscopy. Note that different time (5 and 10 min) and number of

indentations (1–3) were performed as we could only do two preliminary experiments before the machine experienced a technical issue and was no longer usable. All replicates are shown in Appendix Fig. S3.

## Development of a Finite Element Model of the seed

The description of the model is provided in Appendix Supplementary Methods: Model and is available at https://gitlab.inria.fr/mosaic/publications/bauer_etal_simu_supp/.

## Statistical analysis and reproducibility

We did not carry out any sample size estimation before carrying out the experiments but tried to do experiments with the highest number of technical, biological, and experimental replicates given time, money, and space limitations. Seeds from at least three different fruits from three different plants were analyzed per experiment and, when applicable, were randomly distributed among conditions. No blinding was done. Seeds that were too injured during preparation were removed from the analysis. All experiments except the measurements of the shape of mature seeds in the different CMT reporters and the analysis of the cell wall thickness by TEM, have been carried at least two times independently and the resulting data have been pooled. The plots were generated using the R software, Python or Excel. When data are compared using Student tests, statistical significance was displayed with stars as followed: $*P < 0.05$, $**P < 0.01$, $***P < 0.001$ and $****P < 0.0001$. In the boxplot representations, the midline represents the median of the data, while the lower and upper limits of the box represent the first and third quartile, respectively. The error bars represent the distance between the median and one and a half time the interquartile range. When possible, individual values, corresponding to single seeds, are also overlayed as points. When relevant, violin plots are also displayed to get a better grasp of the shape of the distribution. In the frequency plots, the bold lines show the trending curves and the dotted lines show the mean value when all cells and seeds of a given condition are pooled. The plots showing seed growth rate and seed growth anisotropy were obtained by deriving the daily measurements of seed size and aspect ratio using the following formula:

$$mean_{Growth\ rate_{n \to n+1}} = \frac{(mean_{Area_{n+1}} - mean_{Area_n})}{mean_{Area_n}}$$

The error bars show the standard deviation calculated using the following formula:

$$sd_{Growth\ rate_{n \to n+1}} = \sqrt{\left(\frac{sd_{Area_{n+1}}}{mean_{Area_{n+1}}}\right)^2 + \left(\frac{sd_{Area_n}}{mean_{Area_n}}\right)^2}$$

## Data availability

All the raw images and data shown in this study are deposited in Zenodo and can be downloaded following this link: https://doi.org/10.5281/zenodo.10906791. The code for the computational simulations is available on the following GitHub link: https://gitlab.inria.fr/mosaic/publications/bauer_etal_simu_supp/.

The source data of this paper are collected in the following database record: biostudies:S-SCDT-10_1038-S44318-024-00138-w.

## Peer review information

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

## Acknowledgements

The authors thank Stéphane Verger, Alice Malivert and Olivier Hamant for providing the *ktn1*, *csi1-3*, *prc1-1* and *spr2-2* mutants and the *p35S::TUA6-GFP, pPDF1::mCitrine-MBD and pPDF1::CFP-N7* reporters, Miguel Botella for providing the *ttl1 ttl3* mutant, Frederic Berger for providing the *iku2-2* mutant, Marilyn Vantard for providing the *p35S::MAP65-1-RFP* reporter and Marie-Cecile Caillaud for providing the *MAP65-1-pENTR-L1-L2* plasmid. The authors thank Charlotte Kirchhelle for providing insanely good ideas that substantially improved the quality of this work. The authors also thank Olivier Hamant, Marie-Cécile Caillaud, and Magalie Uyttewaal for helpful discussion and comments on the manuscript; Audrey Creff, Vincent Bayle, Claire Lionnet, and Corentin Mollier for technical assistance with experiments and analysis. The authors also thank Jeremy Just for his help in resolving computer issues. We acknowledge the contribution of SFR Biosciences (Universite Claude Bernard Lyon 1, CNRS UAR3444, Inserm US8, ENS de Lyon) PLATIM-LyMIC, and the Bio21 Advanced Microscopy Facility (University of Melbourne) for technical assistance with microscopy; Alexis Lacroix, Patrice Bolland, Camille Knaupp and Justin Berger for technical assistance with plant cultivation; Isabelle Desbouchages and Hervé Leyral for technical assistance regarding molecular biology work; Cindy Vial, Laureen Grangier, Nelly Camilleri, Stéphanie Maurin and Julie Prata for administrative assistance. The PhD thesis of Amélie Bauer was supported by a joint PhD program between the CNRS and the University of Melbourne. The PhD thesis of Camille Bied was supported by a fellowship from the French Ministry of Higher Education. This work is also supported by the French National Agency of Research (ANR, grant agreement ANR-23-CE13-0009, "Mechaseed") and by the European Research Council (ERC, grant agreement No 101019515, "Musix").

## Author contributions

**Amélie Bauer**: Conceptualization; Resources; Validation; Investigation; Visualization; Writing—review and editing. **Olivier Ali**: Conceptualization; Data curation; Software; Formal analysis; Methodology. **Camille Bied**: Investigation. **Sophie Boeuf**: Investigation. **Simone Bovio**: Investigation. **Adrien Delattre**: Investigation. **Gwyneth Ingram**: Conceptualization; Supervision; Funding acquisition; Investigation; Writing—review and editing. **John F Golz**: Conceptualization; Supervision; Funding acquisition; Investigation; Writing—review and editing. **Benoit Landrein**: Conceptualization; Data curation; Formal analysis; Supervision; Funding acquisition; Validation; Investigation; Visualization; Methodology; Writing—original draft; Project administration; Writing—review and editing.

Source data underlying figure panels in this paper may have individual authorship assigned. Where available, figure panel/source data authorship is listed in the following database record: biostudies:S-SCDT-10_1038-S44318-024-00138-w.

## Disclosure and competing interests statement

The authors declare no competing interests.

# Expanded View Figures

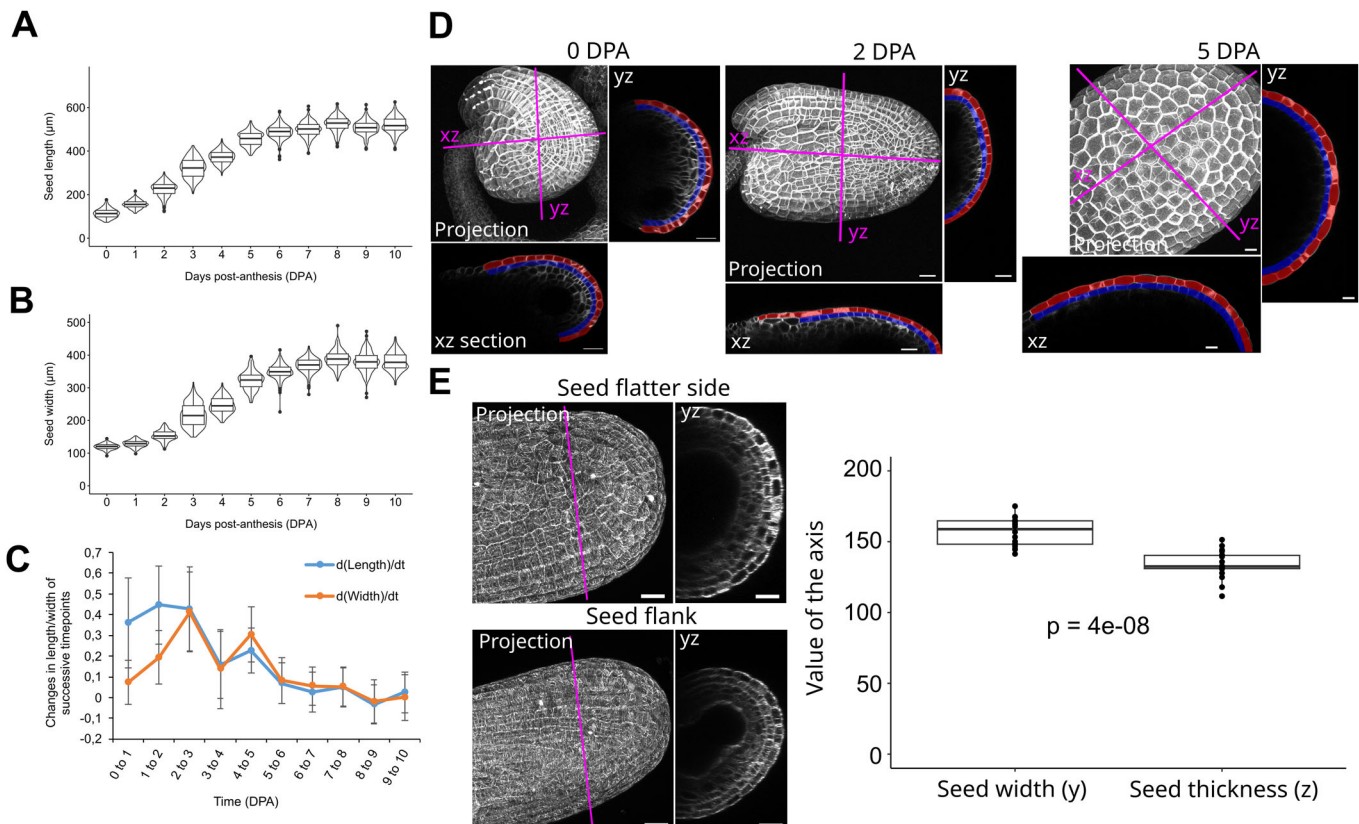

**Figure EV1. Evolution of the width, length and height of developing seeds.**

(**A, B**) Measurements of the length (**A**) and width (**B**) of WT seeds (Col-0 ecotype) from 0 to 10 days post anthesis (10DPA), $n = 180$–209 seeds per day, two independent experiments. In the boxplot representations, the midline represents the median of the data while the lower and upper limits of the box represent the first and third quartile, respectively. The error bars represent the distance between the median and one and a half time the interquartile range. (**C**) Relative changes in seed length and width over time obtained by deriving the measurements of seed length and width of (**A, B**). The error bars show the standard deviation of the derivative (see "Methods"). (**D**) Representative z-projection and middle sections along the width of the seed imaged using the ubiquitous membrane marker (*p35S::LTi6b-GFP*) at 0, 2 and 5DPA. The outer integument abaxial and adaxial epidermis are overlayed in red and blue, respectively. Scale bars, 20 μm. (**E**) Comparison between seed width and seed thickness obtained by manually fitting an ellipse on the seed surface of middle sections of seeds oriented on their flatter sides or on their flank, $n = 8$ to 10 seeds, two independent experiments. Scale bars, 20 μm. Data were compared using bilateral Student tests. In the boxplot representations, the midline represents the median of the data while the lower and upper limits of the box represent the first and third quartile, respectively. The error bars represent the distance between the median and one and a half time the interquartile range.

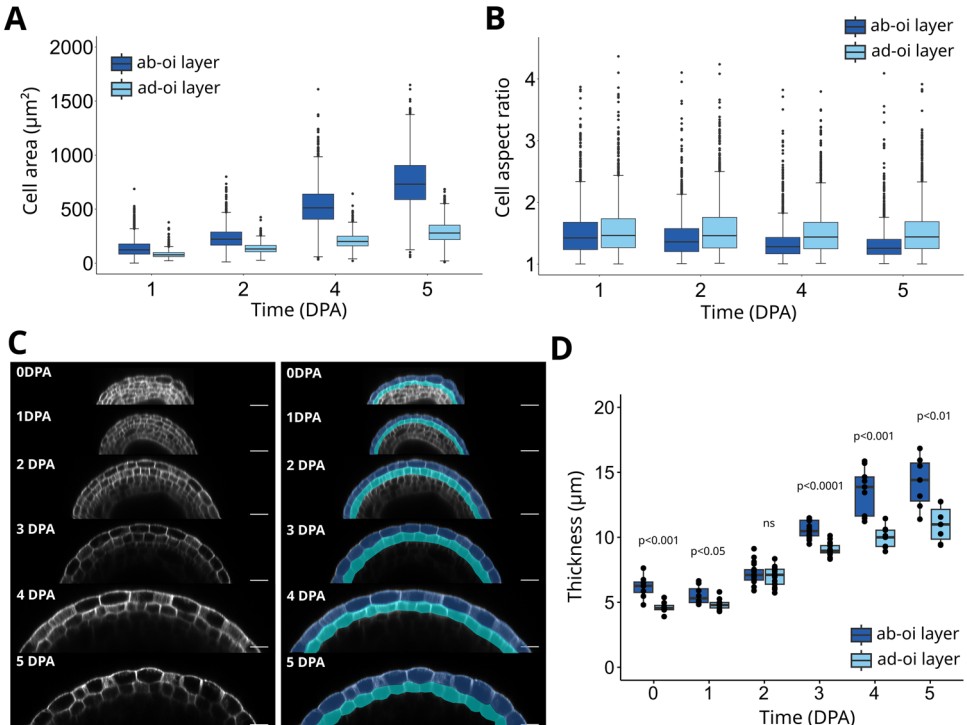

**Figure EV2. Measurements of outer integument cell size and shape.**

(A, B) Evolution of the size (measured as area) and shape (measured using the aspect ratio) of the cells in the abaxial and adaxial epidermis of the outer integument (DPA: Days post anthesis), 1373 to 2402 cells from 10 to 11 seeds, two independent experiments. In the boxplot representations, the midline represents the median of the data while the lower and upper limits of the box represent the first and third quartile, respectively. The error bars represent the distance between the median and one and a half time the interquartile range. (C) Representative middle sections along the width of the seeds (imaged using the microtubule reporter *p35S::MAP65-1-RFP*) showing the evolution of the thickness of the cells in the abaxial (overlayer in dark blue) and the adaxial (overlayed in light blue) epidermis of the outer integument. Scale bars: 20 μm. (D) Evolution of the thickness of the cells in the outer integument abaxial (ab-oi) and adaxial (ad-oi) epidermis from 0 to 5DPA based on manual measurements performed on sections similar to the ones presented in (C), $n = 45$–60 cells from 9 to 12 seeds, two independent experiments for 0, 1, 2, 3, 5,DPA, one experiment for 4DPA. Data were compared using bilateral Student tests. In the boxplot representations, the midline represents the median of the data while the lower and upper limits of the box represent the first and third quartile, respectively. The error bars represent the distance between the median and one and a half time the interquartile range.

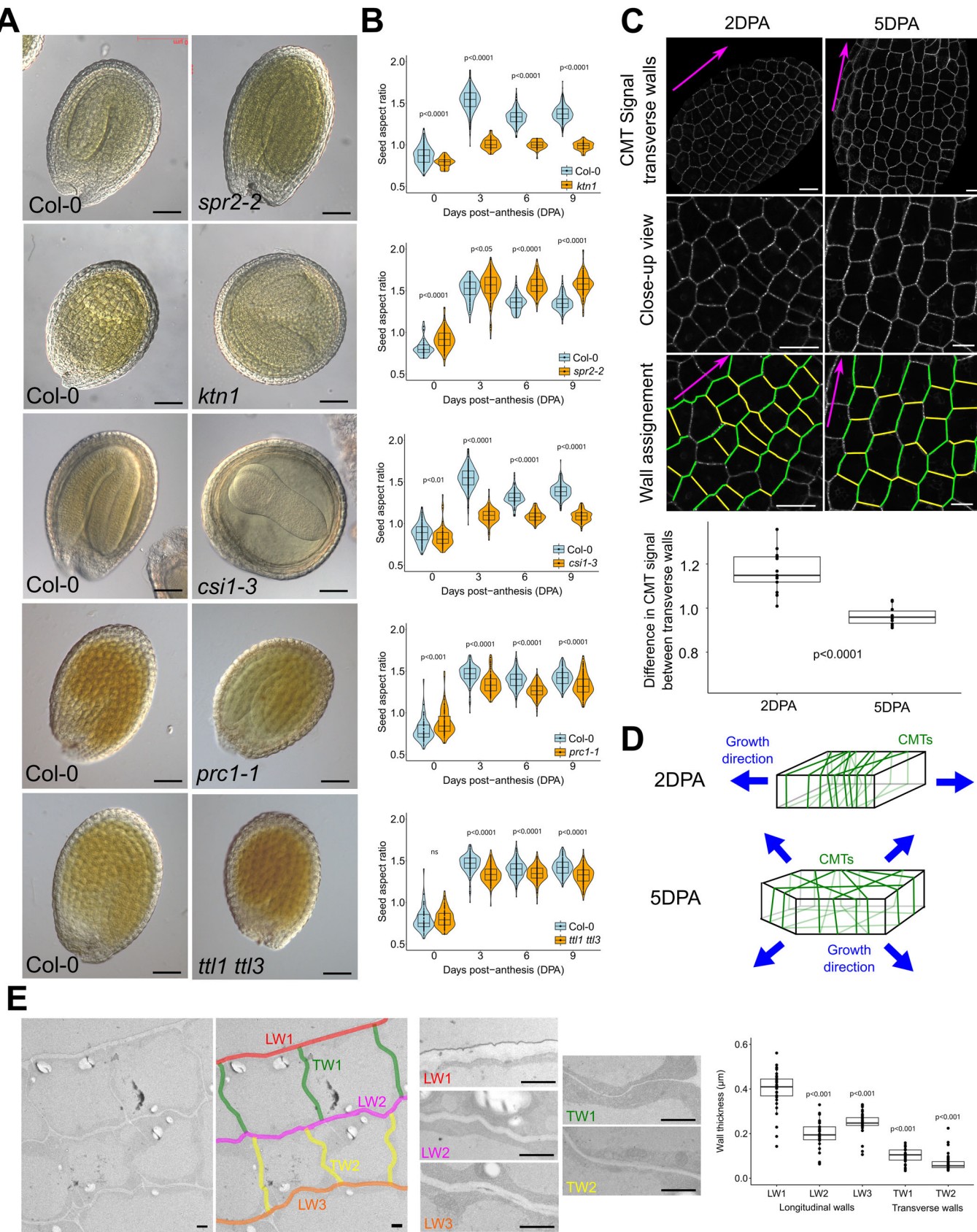

◀ **Figure EV3.   Mechanical control of seed elongation.**

(A) Seed shape phenotype at 9DPA of the mutants of CMT organization and response to forces: *katanin* (*ktn1*) and *spiral2* (*spr2*), the mutants of cellulose guidance by the CMTs: *csi1-3* and *ttl1 ttl3*, and the mutant of cellulose synthase subunit: *prc1-1* (*cesa6*). Scale bars: 100 µm. (B) Evolution of the aspect ratio of developing WT, *ktn1*, *spr2-2*, *csi1-3, ttl1 ttl3*, and *prc1-1* (*cesa6*) and mutant seeds, $n = 28–366$ seeds per day per genotype, two independent experiments. Data were compared using bilateral Student tests. In the boxplot representations, the midline represents the median of the data while the lower and upper limits of the box represent the first and third quartile, respectively. The error bars represent the distance between the median and one and a half time the interquartile range. (C) Quantification of the CMT signal facing transverse walls in the seed coat outer integument abaxial epidermis (imaged using the *p35S::MAP65-1-RFP* reporter) as a function of the orientation of the wall relative to the main seed axis (perpendicular or parallel), $n = 100–130$ walls from 10 to 13 seeds, two independent experiments. Data were compared using a bilateral Student test. Scale bars, 10 µm. In the boxplot representations, the midline represents the median of the data while the lower and upper limits of the box represent the first and third quartile, respectively. The error bars represent the distance between the median and one and a half time the interquartile range. (D) Model of the organization of the CMTs in the outer integument abaxial epidermis at 2DPA and 5DPA. (E) TEM iImaging and quantification of the mean thickness of transverse and longitudinal walls of outer integument cells of seeds at 3DPA, $n = 32–42$ walls from 5 seeds, one experiment. All walls were compared to wall 1 using bilateral Student tests. Scale bars, 2 µm. In the boxplot representations, the midline represents the median of the data while the lower and upper limits of the box represent the first and third quartile, respectively. The error bars represent the distance between the median and one and a half time the interquartile range.

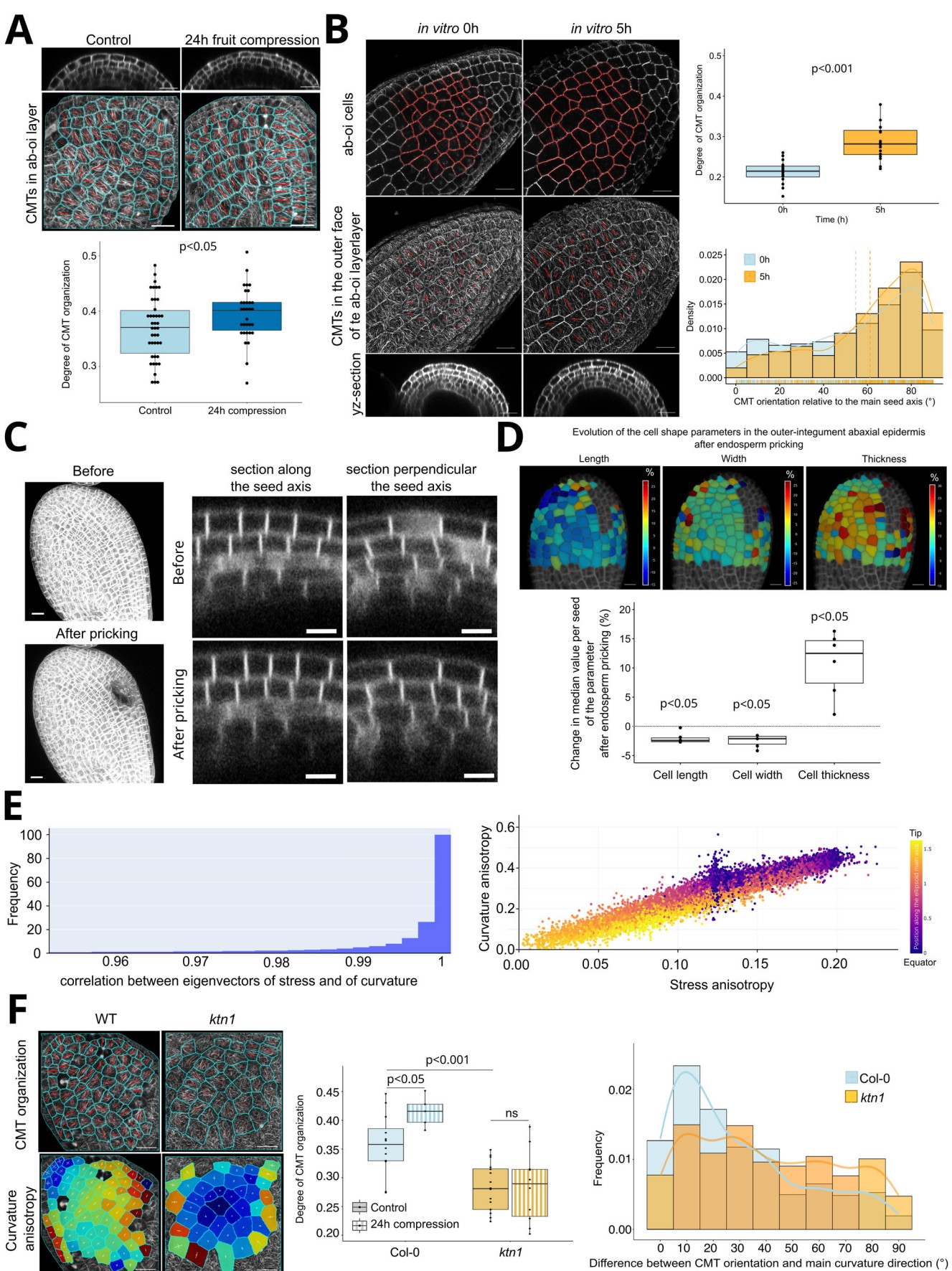

◀ **Figure EV4. CMTs in the outer face of the outer integument abaxial epidermis align according to shape-driven stresses.**

(A) Organization of the CMTs (imaged using the *p35S::MAP65-1-RFP* reporter) in the outer face of abaxial outer integument epidermis during the anisotropic growth phase (2DPA) in control seeds or in seeds whose fruits were compressed for 24 h with a microvice (following the protocol of Creff et al, 2015), $n = 1512–2319$ cells from 23 to 32 seeds, five independent experiments. In the pictures, the orientation of the red bars shows the mean orientation of the CMTs in each cell and its length shows its degree of organization. Data were compared using a bilateral Student test. In the boxplot representations, the midline represents the median of the data while the lower and upper limits of the box represent the first and third quartile respectively. The error bars represent the distance between the median and one and a half time the interquartile range. (B) Effect of a 5 h cultivation of developing seeds at 2DPA in liquid culture medium on the growth (representative pictures) and the organization of the CMTs facing the outer face (representative pictures and quantification) of the cells of the outer integument abaxial epidermis, $n = 741–751$ cells from 16 seeds, two independent experiments. Data were compared using a bilateral Student test. Scale bars, 10 μm. In the boxplot representations, the midline represents the median of the data while the lower and upper limits of the box represent the first and third quartile, respectively. The error bars represent the distance between the median and one and a half time the interquartile range. (C) Representative z-projections and sections parallel and perpendicular to the main seed axis of 2DPA seeds (imaged using the ubiquitous membrane marker *LTi6b-GFP*) showing the effect of a release of endosperm pressure by pricking on the aspect of seed coat cells. Scale bars, projections: $n = 20$ μm, sections: 10 μm. (D) Quantification of the median changes in length, width and height of the cells in the outer integument abaxial epidermis following endosperm pricking (see "Methods"), $n = 669$ cells from 6 seeds, two independent experiments. Scale bars, 10 μm. Data were compared to 0 using Wilcoxon tests. In the boxplot representations, the midline represents the median of the data while the lower and upper limits of the box represent the first and third quartile, respectively. The error bars represent the distance between the median and one and a half time the interquartile range. (E) Correlation between main stress direction and main curvature direction, and between stress anisotropy and curvature anisotropy based on the computational simulations shown in Fig. 4C and described in Appendix Supplementary Methods. (F) Quantification of the correlation between the main orientation of the CMTs and the main direction of curvature, and of the degree of organization of the CMTs following a 24 h compression, of WT and *ktn1* seeds at 2DPA. In the CMT pictures, the orientation of the red bars shows the mean orientation of the CMTs in each cell and their length, their degree of organization. In the heatmaps of curvature anisotropy, the orientations of the two perpendicular white bars represent the axes of maximum and minimum curvature in each cell and their lengths the degree of curvature in each of these two directions. Data were compared using bilateral Student tests, Curvature maps: Col-0: $n = 407–763$ cells from 6 to 14 seeds, *ktn1*: $n = 437–1008$ cells from 10 to 17 seeds, four independent experiments; compressions: Col-0 : $n = 723$ cells from 13 seeds, *ktn1* : $n = 934$ cells from 15 seeds, four independent experiments. Scale bars: 20 μm. In the boxplot representations, the midline represents the median of the data while the lower and upper limits of the box represent the first and third quartile, respectively. The error bars represent the distance between the median and one and a half time the interquartile range.

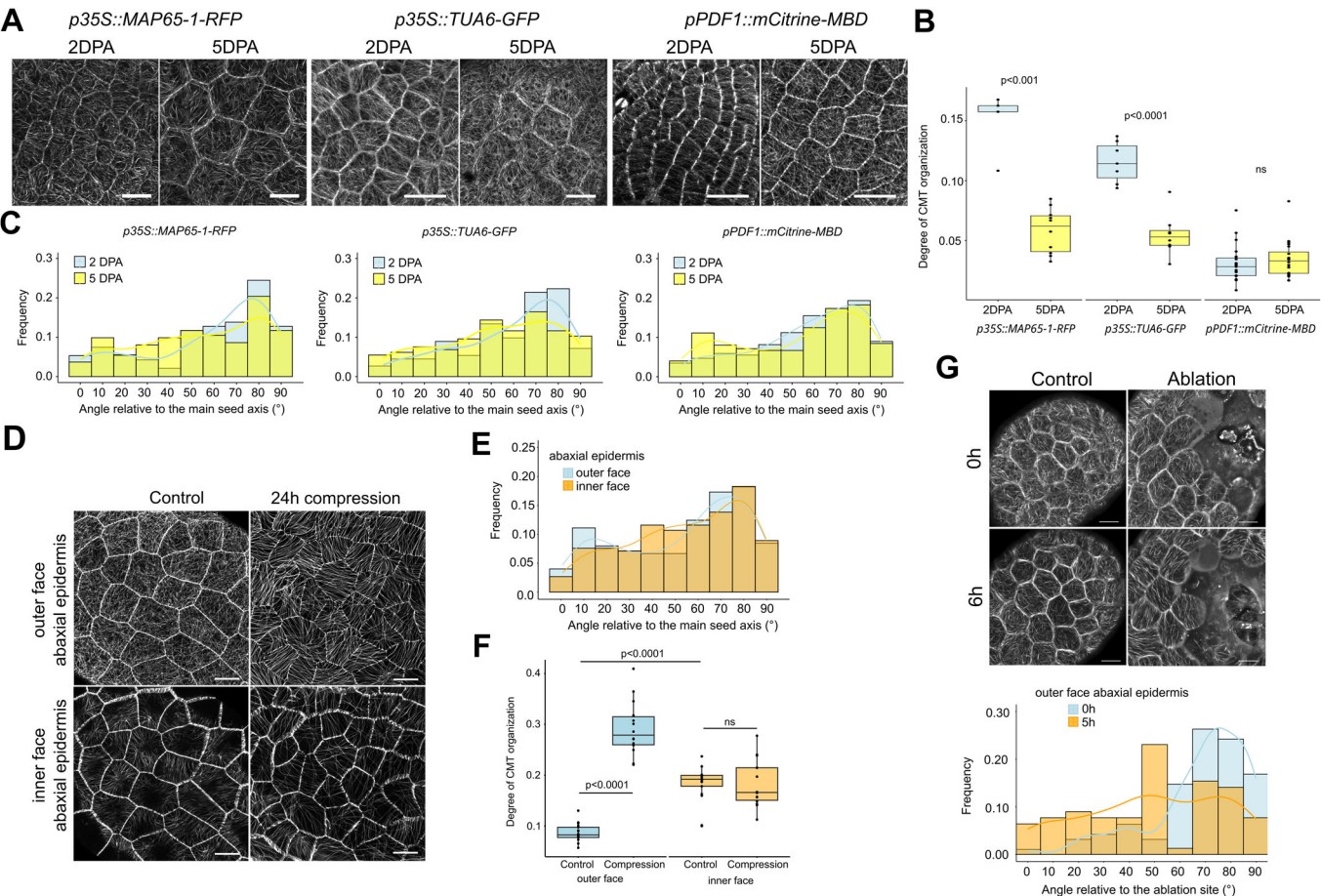

**Figure EV5. CMT organization and response to forces in the outer integument abaxial epidermis during the isotropic growth phase.**

(A–C) Comparison of the organization and orientation relative to the main seed axis of the CMTs in the outer face of the abaxial outer integument epidermis during the anisotropic growth phase (2DPA) and during the isotropic growth phase (5DPA) imaged using three different CMT reporters (*p35S::MAP65-1-RFP*, *p35S::TUA6-GFP*, *pPDF1::mCitrine-MBD*), scale bars: 20 μm, n = 94–491 cells from 5 to 23 seeds, two to three independent experiments depending on the reporter. Data were compared using bilateral Student tests. In the boxplot representations, the midline represents the median of the data while the lower and upper limits of the box represent the first and third quartile, respectively. The error bars represent the distance between the median and one and a half time the interquartile range. (D) Organization of the CMTs array facing the inner of the outer face of the seed in the abaxial epidermis at 5DPA (imaged using the *pPDF1::mCitrine-MBD* reporter) following a 24 h compression of the seed. Scale bars: 20 μm. (E) Quantification of the orientation relative to the main seed axis of the CMTs facing the inner and the outer face of the seed in the abaxial epidermis at 5DPA (imaged using the *pPDF1::mCitrine-MBD* reporter), n = 273 cells from 18 seeds. (F) Effect of a 24 h compression of the fruit on the degree of organization of the CMTs (imaged using the *pPDF1::mCitrine-MBD* reporter) facing the inner or the outer side of the seed in the abaxial outer integument epidermis at 5DPA, n = 273–293 cells from 14 to 18 seeds, two independent experiments. Data were compared using bilateral Student tests. In the boxplot representations, the midline represents the median of the data while the lower and upper limits of the box represent the first and third quartile, respectively. The error bars represent the distance between the median and one and a half time the interquartile range. (G) Effect of cell ablations on the orientation of the CMTs (imaged using the *p35S::MAP65-1-RFP* reporter) facing the outer face of 5DPA seeds in the abaxial outer integument epidermis, n = 78–95 cells of 12 to 13 seeds, scale bars: 20 μm, three independent experiments.

