## [Peer Review File · The EMBO Journal]

Spatiotemporally distinct responses to mechanical forces shape the developing seed of Arabidopsis

Amélie Bauer, Olivier Ali, Camille Bied, Sophy Boeuf, Simone Bovio, Adrien Delattre, Gwyneth Ingram, John Golz, and Benoit Landrein

Corresponding author: Benoit Landrein (benoit.landrein@ens-lyon.fr)

Review Timeline:

Submission Date:	23rd Aug 23
Editorial Decision:	26th Sep 23
Appeal:	2nd Oct 23
Editorial Decision:	25th Oct 23
Revision Received:	2nd Apr 24
Editorial Decision:	24th Apr 24
Revision Received:	6th May 24
Accepted:	22nd May 24

Editor: Ieva Gailite

Transaction Report:

Dear Dr. Landrein,

Thank you for submitting your manuscript for consideration by The EMBO Journal. We have now received three reviewer reports on your manuscript, which are included below for your information. Based on these comments, we unfortunately had to conclude that the study is not a sufficiently strong candidate for publication in The EMBO Journal.

As you can see, reviewer #2 indicates a number of scholarly issues, as well as technical and data interpretation concerns that affect the conclusions of the manuscript. In addition, reviewer #3 finds that the conclusions are not sufficiently supported and raises concerns with the quality of the images used for microtubule quantification. While reviewer #1 is more positive in their comments, they also indicate a number of limitations regarding the analysis and highlights its correlative nature. Based on these assessments, we unfortunately cannot offer to invite a revised manuscript at The EMBO Journal, as we require a strong support from the majority of the referees for further consideration here.

While we cannot pursue this manuscript further, I would like to suggest a transfer to our not-for-profit open-access sister journal, Life Science Alliance (LSA). I have shared your manuscript and the accompanying reviews with LSA Executive Editor, Eric Sawey, who is interested in these findings, and would like to invite further consideration of this manuscript at LSA pending the following revisions:

- Address Reviewer 1's comments.
- Address Reviewer 2's comments via added discussion and clarification. Certain claims should be tone down and presented with alternative explanations in response to the concern that there is not enough evidence to conclude that microtubules are causative for anisotropic growth.
- Address Reviewer 3's comments.

We understand that such a revision might need to be re-reviewed, in which case, Dr. Sawey will walk the Reviewers through our transfer process. If you are interested in this option, please use the link below to transfer your manuscript to LSA:

Link Not Available

You do not need to revise the manuscript before transferring it to LSA. Once you transfer, Dr. Sawey will email you an invitation to revise and resubmit, listing the same revision requests as mentioned above. Please feel free to reach out at e.sawey@life-science-alliance.org if you have any questions about the LSA journal, the transfer process or the revisions requested.

Thank you in any case for the opportunity to consider this manuscript. I am sorry that I could not offer better news this time, but I nevertheless hope that you will find the transfer offer of interest.

Yours sincerely,

Ieva Gailite

Referee #1:

A very interesting, well written, and comprehensive study on the biomechanics of seed development. The way the results are presented, starting from the morphological features of growing seeds, to cellular growth differentials to a subcellular picture based on microtubule-reorientations, is appealing and I have enjoyed reading this manuscript. Even though the role of mechanics in morphogenetic processes in plants has been explored in a variety of organs, this study focusing on seed development is novel and timely. Moreover, the experiments are carefully described and conducted and support the conclusions of the paper. As such, this papers meet all requirements per the EMBO journal guidelines.

I have some small remarks/questions but these should be considered as minor:

- The authors start their story with a morphological characterization of seeds during their growth, revealing distinct stages. However, this is all based on a 2-dimensional measurement of an obviously three-dimensional object. Why was the third dimension not considered? Is it possible that taking this into consideration would have revealed additional stages, or is there some reason (unbeknownst to this reviewer) why the chosen two-dimensional projection is the only relevant one?
- In their second section "Seed shape depends on..." Of the results and discussion the authors refer to previous work in stating that seed size is the product of a mechanical balance between endosperm and seed coat. It made me wonder what the role of the embryo inside the seed is, which grows and develops in tandem; would a growing embryo not also provide an additional internal pressure in the seed whose action must be balanced at the exterior to reach a stable mechanical object?
- In the same section, the authors remark that their data suggests that seed shape is dictated by the seed coat mechanics. Given that shape (geometry) in growing solid objects is usually the product of two (or more) opposing mechanical forces so that these can be balanced to produce a stable final state, this comments seems somewhat counter-intuitive to me, and may require elaboration.
- Perhaps nit-picking (given the wealth of literature on this topic in other tissues) but the authors experimentally establish a correlation between CMT alignment and the direction of anisotropic growth (The section starting with "We used the Fibrtool plugin"). Then they conclude a causality from this correlation in that CMT orientation determine the axis of seed elongation. Strictly, their data does not show that, it shows that the two are correlated, but this does not rule out that the anisotropic growth orients the CMTs, and not the other way around. Also in the subsequent paragraph a similar observation can be made about correlation vs causation.
- In the description of the study of MT mutants (*ktn1*, *spr2*, etc), the authors provide the names of the mutated genes, and then state that the observed effects are in line with the function of these genes, without making it clear to the reader what these factors are. I think, given that this is an important part of this study, this would be courteous. For example, explain that katanin is a MT severing factor so that a loss-of-function leads to a much less dynamic CMT array which also loses its sensitivity to mechanical stress, and then explain how this links to the observations in this study. The way it is currently written asks the reader (if they are not CMT experts) to go through a lot of literature to interpret some central findings in this paper.
- This paper is good and publishable as is. However, it did make me wonder, similarly as with other papers in the same spirit for other tissue types, if the connections between mechanics>CMTs>morphogenesis could not be study in a more causal way in systems in which CMTs can be disrupted in an inducible manner (e.g. using opto-genetics or chemical induction), so that one could directly manipulate the CMT at a particular phase in such a complex process, and see if the resulting effect is consistent with the hypothesis that is formulated. For example the opto-Katanin method for mammalian cells, would be a fantastic tool, but of course cannot be implemented in plants without re-engineering. I am not suggesting this should be done here, but I think it would be a valuable tool to get a significant step closer to a real proof of the causal relationships that are proposed in this paper, and others.

Referee #2:

In the paper entitled "Spatiotemporally distinct responses to mechanical forces shape the developing seed of *Arabidopsis*" Bauer et al. seek to address the challenging question of how organ-scale control of anisotropy of seeds is controlled. They conduct a nice phenotypic analysis of the shape changes that occur at the organ scale. This is accompanied by cellular scale quantitative analyses of the growth behaviors that occur as seed growth behaviors transition from anisotropic to isotropic. This part of the paper is solid. The remainder of the paper that attempts to conclude that specific arrays of microtubules on specific faces of the seed epidermal cells sense mechanical stress to orchestrate the morphogenesis process. Overall their conclusions are not supported by the data or by the cited prior publications that are used to justify the overall approach. Major weaknesses are explained below.

The literature base in the field of mechanosensing is not covered in an accurate or complete manner. This pertains to the field in general and the specific functions of *KATANIN* and *SPIRAL2* in this context. These errors lead to a failure to define the true knowledge gaps, a failure to justify their approach, and widespread over-interpretation of the data in this paper. These weaknesses in scholarship confuse the reader, hinders advancement of the field, and leads to reduced credibility. Several examples are provided below with the goal of helping the authors improve the paper.

Scholarship issues

1) The authors state:

"Mechanical forces have notably been shown to affect CMT organization, and thus cellulose deposition (Hamant et al, 2008; Sampathkumar et al, 2014; Robinson & Kuhlemeier, 2018; Hejnowicz et al, 2000), so that shape-driven stresses can affect anisotropic growth in a variety of plant organs (Trinh et al, 2021)."

These papers in isolation or in aggregate do not make clear connections between stress, microtubules, cellulose, and growth patterns. The authors fail to cite more recent papers that make mechanistic contributions in this area.

2) The authors state:

"In many aerial organs, it is believed that growth is promoted by the pressure of inner tissues but restricted by the mechanical properties of one or several outer layers, which is often the epidermis (Kutschera & Niklas, 2007)."

This ignores turgor-generated forces in the epidermis that contribute to expansion of the tissue and the organ. The authors fail to account for recent papers that analyze in detail the relationships between stress, microtubules, and cell wall anisotropy in the epidermis that can strongly influence the growth of the tissue.

Zhao, F., Du, F., Oliveri, H., Zhou, L., Ali, O., Chen, W., Feng, S., Wang, Q., Lu, S., Long, M., et al. (2020). Microtubule-Mediated Wall Anisotropy Contributes to Leaf Blade Flattening. *Curr Biol* 30, 3972-3985 e3976.

Belteton, S., Li, W., Hatam, F.A., Quinn, M.I., Szymanski, M.R., Marley, M., Turner, J.A., and Szymanski, D.B. (2021). Real-time conversion of tissue-scale mechanical forces into an interdigitated growth pattern. *Nature Plants* 7, 826-841.

Schneider, R., Ehrhardt, D.W., Meyerowitz, E.M., and Sampathkumar, A. (2022). Tethering of cellulose synthase to microtubules dampens mechano-induced cytoskeletal organization in Arabidopsis pavement cells. *Nat Plants* 8, 1064-1073.

3) The authors state:

"In pavement cells, differences in mechanical properties between contiguous walls, also associated with local pectin accumulation, trigger the initiation of the lobes through buckling, which are then amplified through CMT responses to forces (Majda et al, 2017; Sampathkumar et al, 2014)."

Although not directly related to the topic of organ-scale morphogenesis these statements inaccurately portray the field of stress sensing and morphogenesis. The papers cited below do not prove the points the authors are making. The role of pectin in lobe formation is not supported by clear experimental data and should not be used as an illustrative example.

4) In the results section, the authors state:

"Given that this latter class of seeds have phenotypically wildtype endosperm and embryos (being heterozygous), but a phenotypically mutant seed coat owing to its maternal origin (Fig 1E), our data suggest that seed shape mainly depend on the mechanical properties of the seed coat."

The involvement of KTN1 and SPR2 in stress sensing is vastly overstated and this leads to an overinterpretation of the mutant phenotypes in this paper. This is a major problem. The phenotypic data do not say anything direct about the mechanical properties of the seed coat. It just means maternal tissues require wild type KTN and SPR2. The connections between *ktn* and *spr2* phenotypes and stress sensing are not clearly proven in this paper (see below).

Technical issues

1) The paper claims to link stress, microtubule patterns, and anisotropic growth. In the experimental system the stress patterns are not known, neither is the critical cell face of the cell known. The authors state:

"CMTs determine the main axis of cell elongation by controlling the oriented deposition of cellulose microfibrils in the wall (Paredes et al, 2006). In leaves, sepals and meristems, the CMTs facing the outermost cell wall of the epidermis, which is believed to be load-bearing, organize according to shape-driven stresses and are believed to control the anisotropic growth of these organs (Hamant et al, 2008; Hervieux et al, 2016; Sampathkumar et al, 2014; Zhao et al, 2020). However, in hypocotyls, it is the CMTs that face the inner side of the epidermis that are robustly oriented perpendicularly to the main axis of elongation (Crowell et al, 2011; Chan et al, 2011). This suggests that the inner wall of the epidermis is load-bearing and controls the elongation of this organ. This scenario is also consistent with computational simulations and experiments on hypocotyls (Robinson & Kuhlemeier, 2018; Verger et al, 2018) (Robinson 2018, Verger 2018)."

The authors rely on a cellular scale anisotropy measurements and an assumed correlation with stress to conclude which cell face mediates stress dependent anisotropic growth. First, it may not be stress dependent control. There are many mechanisms by which organ scale shape change can occur. The authors rely on a superficial correlation between microtubule order and an assumed stress pattern based on cell curvature to make their conclusions. If the process is completely microtubule-mediated (not shown here) the authors really don't know what microtubule array is responsible. They ignore the possible contribution of anisotropy in the anticlinal walls, which is known to affect anisotropic growth in leaves. In addition, the authors have no clear idea of the stress patterns in the cells/tissue. They assume that stress is related to local curvature, but many other factors besides local curvature could define the stress patterns in the cell. Curvature just correlates with the microtubule pattern. It is also unclear what the stress patterns of the seed will be upon squeezing. There are likely combinations of compression and tensile forces that change upon squeezing and this is impossible to intuit. Some kind of reliable FE model would be needed to even attempt a guess.

2) The *ktn* and *spr2* mutant analyses cannot be linked to stress sensing and microtubule reorganization because there are no strong published data indicating that either of these genes function in that capacity. This point is alluded to above. In the absence of such data, either explained clearly based on a published result or new data, the interpretation of the mutant phenotypes is flawed.

Referee #3:

In this study, Bauer et al investigate the development of seeds of Arabidopsis in context of growth trajectories and mechanobiology.

The authors use growth series of Arabidopsis seeds to conclude that growth follows an early anisotropic phase and then switches to an isotropic phase. They show that the first phase is accompanied and largely dependent on microtubule

organization in the outer cell layers of the seed coat and that the second growth phase is largely dependent on inner cell layers of the seed coat and the microtubule organization in these layers. The work is largely well conducted and logically written. While I do find the study compelling, I have some comments that I think is necessary for both this study and the field of mechanobiology and the corresponding microtubule organization analyses in general.

Major points:

While I applaud the authors for using multiple fluorescence microtubule marker lines to analyze and quantify microtubule organization, I am still left with a sense of uncertainty regarding the conclusions. This sense is actually something I have had in context to several other studies of microtubule quantification with the fibril tool and stress applications. In the different figures of microtubule fluorescence, I am unable to see any real differences. This is even in the blow-ups of images I obtained from the EMBO Journal for reviewing purposes. I am of course aware of the quantification that is associated with this, but I would feel a bit uncomfortable to support conclusions based on the images that are supplied. For example, Figure 3: I cannot say that the images here really support the claims as I cannot see really clear microtubules at least in the outer face and inner face of the adaxial epidermis. In addition, the microtubule alignments appear high in the abaxial outer face cells, but many of these are not aligned with the growth axis, and so this is a bit troubling. I still have to point out that I do not doubt the conclusions of the paper and I think the authors have done a better job than many other studies in this field. I am just saying that with these types of images, I am not confident about quantification and how well a fibril tool does. I will leave it up to the authors to argue for the use of these or not. In this context, I would like the authors to zoom in and convince the readers that the microtubules can clearly be seen in cells that they use for quantifications.

Minor points:

The authors may want to exchange some images as for example the Figure 6, EV3 and EV6 contain the exact same images. This is not indicated and I would assume the authors have more than these images to convince the reader (please recheck the rest of your figures to avoid any such duplications). If published this could have some rather unwanted issues.

Some spelling mistakes in figures need to be corrected (for example Anisotropic in Fig 2).

Go through the manuscript and check some spelling etc...for example MAP61 instead of MAP65 etc.

The reduction of cellulose and MT organization virtually always result in loss of anisotropic growth, so not sure how much value this really is to the study?

The authors talk about isotropic microtubule array in cells facing the inner side of the seeds during the isotropic growth face, but then say that they are somewhat longitudinally organized. These statements are in contrast to each other.

The authors talk about "slightly less" preferentially organized...what does this mean??

** As a service to authors, EMBO Press provides authors with the possibility to transfer a manuscript that one journal cannot offer to publish to another EMBO publication or the open access journal Life Science Alliance launched in partnership between EMBO Press, Rockefeller University Press and Cold Spring Harbor Laboratory Press. The full manuscript and if applicable, reviewers' reports, are automatically sent to the receiving journal to allow for fast handling and a prompt decision on your manuscript. For more details of this service, and to transfer your manuscript please click on Link Not Available. **

Dear editor,

I am coming back to you following the recent review of our manuscript in Embo Journal. Your decision to reject our paper seems mainly motivated by the very negative report of reviewer 2, although reviewers 1 and 3 appear to be quite positive about the manuscript (both stating that they would see it fit well in Embo Journal once we have corrected some issues). However, after carefully studying the arguments of reviewer 2, I feel that I can argue about the objectivity of this reviewer. It seems to me that he is not criticizing our paper, but that he is either overlooking or simply refusing to consider the published literature supporting the contribution of shape-driven stresses on microtubule organization. This is notably seen in its critics of the use of the /katanin /and /spr2/ mutants (the strongest issue he has with our work) which, he states, is "vastly overstated". Given all of the articles I list below showing a contribution of either KATANIN, SPR2 or of both proteins to the response to mechanical forces (the two last articles, when taken together, even proposing a molecular mechanism for mechanotransduction and activation of KATANIN upon stress), I really do not see how the reviewer can say that our claim that KATANIN and SPR2 are involved in microtubule response to forces is "vastly overstated" and why we can not use these mutants, as many others have done before:

-Uyttewaal et al, 2012, /Cell/. Mechanical stress acts via KATANIN to amplify differences in growth rate between adjacent cells in Arabidopsis.

-Eng et /al, /2021, /Current Biology/. KATANIN and CLASP function at different spatial scales to mediate microtubule response to mechanical stress in Arabidopsis cotyledons.

-Hervieux /et /al, 2017, /Current Biology/. Mechanical Shielding of Rapidly Growing Cells Buffers Growth Heterogeneity and Contributes to Organ Shape Reproducibility.

-Hervieux /et /al, 2016, /Current /Biology. A Mechanical Feedback Restricts Sepal Growth and Shape in Arabidopsis

-Wightman /et al/, 2013, /Current Biology/. SPIRAL2 determines plant microtubule organization by modulating microtubule severing

-Lin /et al/, 2013, /Current Biology/, Rho GTPase signaling activates microtubule severing to promote microtubule ordering in Arabidopsis

TOGETHER WITH Tang /et al/, 2022, Current Biology, Mechano-transduction via the pectin-FERONIA complex activates ROP6 GTPase signaling in Arabidopsis pavement cell morphogenesis

Similarly, the reviewer also questions our claim that the seed coat of the seed is put under tension by the pressure of the inner endosperm (thus leading to shape-driven stresses), because in pavement cells, the cells are put under tension by inner tissues but also by their own pressure. I do not see, as the reviewer states, how this work in pavement cells is questioning our results given the differences between the seed and the pavement cells (that the reviewer seems to understand very well but to extrapolate too much to other plant organs). Indeed, previous work in the seed cited in our paper (Beauzamy /et al/, 2016 and Creff, Ali /et al/, 2023), together with our analysis of the /iku2/ mutant in this manuscript, clearly support that seed growth is promoted by endosperm pressure, which is putting the seed coat under tension, thus leading to shape-driven stresses. These studies show that the seed is more similar to a meristem in term of mechanics than to a pavement cell. We could discuss the differences between our system and the pavement cells at the end of our manuscript but I really do not see, as the reviewer states, how our results are contradictory with the literature he added in his review. The same applies to his critics of the fact that we did not look at transverse walls, we cannot exclude that they play a role in seed growth control, but it seems reasonable to look at longitudinal walls for organs following the epidermal growth control theory as many studies (including the ones I cited before) have shown a preponderant contribution of these walls to organ growth control.

In summary, I believe that we could answer to the critics of reviewer 2 by pointing better the differences between our system and the pavement cell, and by discussing more our results and their limitations. We can also provide a model showing the pattern of forces in the outer layer of developing seeds to really support that our correlation between microtubule organization and seed curvature is a strong support for a contribution to shape-driven stresses to microtubule organization in the seed. Also I do not think it is strictly necessary, we can also provide evidence that microtubule response to forces in the seed is altered in /ktn1 /and in /spr2 /(as already shown in other organs). I also believe that we can respond to the comments of reviewer 3 about the quality of the images of microtubules by providing high resolution images taken with a confocal microscope equipped with an Airyscan. We can also show a comparative analysis of the influence of the quality of the microtubule image on the quantification with Fibrtool. A minor note as well regarding the point of reviewer 3, the duplication of the panels he mentionned is not a mistake, we are showing

different comparisons between times and between markers in these different figures, it is thus normal that some panels appear multiple times as they correspond to the same condition.

I am sorry to have bothered you with this long argumentation, but I hope that you can, given this response, reconsider your decision to reject our manuscript as I really think that we can answer to the reviewer comments, and that our work, once corrected, would fit very well in Embo Journal.

Best regards,

Benoit Landrein

Revision Plan of the manuscript from Bauer et al.

- Response to the comment of reviewers 1 and 2 regarding the third dimension (i.e. looking at transverse walls):

We have imaged the longitudinal walls of the seed similarly to what was done for many articles looking at the control of organ growth by cortical microtubules, such as: Chan *et al*, 2011; Crowell et al, 2011, Uyttewaal et al, 2012, Hervieux et al, 2016, Robinson et al, 2018, that all show that cell growth anisotropy correlates with CMT organization in longitudinal walls, in agreement with the epidermal growth control theory (Kutschera and Niklas, 2007). However, it is true that Zhao *et al*, 2020 recently showed that transverse microtubules in internal walls also contribute to leaf blade flatening, although it is important to note that the authors also show in their paper that the early elongation of the leaf (or of the sepal) is, like in our case, dependent on CMTs orientation in the outer cell wall of the epidermis.

Given that our confocal images are taken from the top, it would be difficult to study the organization of the microtubules in transverse walls with the resolution that we have. However, we can do two experiments to further support that it is the CMTs in longitudinal walls that control cell growth anisotropy in the abaxial epidermis of the outer-integument:

We can see if there is a **correlation between cell growth direction and CMTs orientation** in the outer face of the abaxial epidermis during the anisotropic growth phase (from 1 to 2 DPA). This can be done in seeds growing in vitro by performing a timelapse with a microtubule reporter (MAP65-1-RFP) and comparing the orientation of the CMTs at 1 DPA with the orientation of cell growth between 1 and 2 DPA.

We can also provide TEM images that we can quantify to show that the outermost cell wall is thicker than any other cell wall (inc. transverse walls) of the seed coat during the anisotropic growth phase. We already have some of these TEM images and just need to have more replicates. This is in agreement with Kierzkowski et al. 2012 that showed that the outer cell wall of the epidermis in the shoot apical meristem is thickened so that it can be load-bearing and withstand the pressure exerted by inner tissues.

- Response to the comment of reviewer 1 regarding the hypothesis that it could be growth anisotropy that orient CMTs and not the opposite like we claim

To answer to this reviewer comment, we can cite the paper from Corson et al, 2009 showing that the growth of the flower buds becomes completely isotropic if we depolymerize microtubules. If this is not enough, **we can apply oryzalin to developing seeds growing in vitro and check that it leads to isotropic growth.** However, given what we know in the literature on the control of cell growth anisotropy by CMTs, I am not sure that this is really needed.

- Response to the comment of reviewer 1 regarding the use of optogenetic tools to alter CMT organization

Unfortunately, we do not have any optogenetic tool available in the lab to alter CMT organization in the seed and I am not sure that they actually exist in plants. However, we can, like stated in the previous point, to **treat developing seeds growing in vitro with oryzalin (to depolymerize the CMTs) and with isoxaben (to block cellulose deposition)** and see how this affects growth anisotropy at cell and organ scale.

- Response to the comment of reviewers 1 and 2 regarding the *katanin* and *spiral2* mutants

Like reviewer 1 proposes, we can **describe better the molecular function of these proteins and what was shown in the literature of their role in the control of microtubule organization, notably in response to forces** (Uyttewaal et al, 2012, Cell, Eng et al, 2021, Current Biology, Hervieux et al, 2017, Hervieux et al, 2016, Wightman et al, 2013, Lin et al, 2013, Tang et al, 2022).

As we have the marker line in the mutant background, we can also **show that the CMTs in the abaxial epidermis are disorganized in the *ktn1* mutant and cannot respond to the application of mechanical forces.** Note that we already performed this experiment once and just need to do an additional replicate.

- Response to the comment or reviewer 2 regarding our assumption that testa walls are put under tension by endosperm pressure (hence leading to the generation of shape driven stresses)

This assumption is supported by the work of Beauzamy et al, 2016 where they used an indenter to measure the overall stiffness of the seed (with large indentations) and showed that at early stages of development, the rigidity of the seed mostly comes from turgor pressure (because it strongly decreases when the seed is plasmolyzed), and more precisely from endosperm pressure (because it also strongly decreases when the seed is pricked to release the pressure of the endosperm). This claim is also supported by our recent paper showing that in the mutant of endosperm development *iku2*, increased endosperm turgor correlates with more testa tension (that is visualized by quantifying the expression of the mechanosensitive gene ELA1, Creff, Ali et al, 2023).

In addition to **describing better these results** (notably the ones of Beauzamy et al, 2016), we can do again some **seed pricking experiments** where we will pierce the seed from the side and see if this releases the tension in the seed coat (imaged from the top with a confocal microscope): Using the membrane marker LTI6b-GFP, we **can see if releasing endosperm pressure leads to a shrinkage of the cells in the abaxial epidermis**, and if this shrinkage occurs similarly in all directions or not, which should not be the case and should further support that cells in the seed coat have anisotropic material properties during the anisotropic growth phase (as described by Kierzkowski et al. 2012). We can also **image CMTs few hours after the release of endosperm pressure by pricking** (provided that the seeds can survive this manipulation), and see if they lose their organization when tension is released.

- Response of the comment of reviewer 3 regarding the quality of the pictures and the use of Fibrtool:

We acknowledge that our pictures were not of good quality because they were compressed too much when the PDF was generated. Our pictures were actually taken on different systems: a Leica SP8, a Nikon and a LSM980. The images taken with LSM980 are of very high quality because this microscope is equipped with the Airy Scan technology. **We can provide high quality images of microtubules taken with each of these microscopes** so that the readers can see well the microtubules and see if their orientation visually correlates with the one calculated with Fibrtool. To further **test how the quality of the images affects the analysis with Fibrtool, we have imaged the same seeds with the SP8 and with the LSM980 and can show, in a supplementary figure, if the mean orientation and degree of organization of the CMTs are the same with both type of acquisition**. According to our preliminary results, it seems that the quality of the image affects the degree of organization that is calculated but not the mean orientation, further supporting that one need to be careful when comparing the degree of organization (as we already described with the different markers). Note that we took that limitation into account in Fig.3 where used the same microscope (LSM980) and MAP65-1 reporters (expressed under the control of different promoters) for our comparison of CMT organization in different walls.

- Response to the comment of reviewer 2 and of the editor regarding the response to forces and its speed

To further test that the CMTs in the abaxial epidermis are organized in response to forces, we propose to do the following experiments where we look at the fast response of CMT to mechanical perturbations:

- As previously described, we can **release endosperm pressure by pricking the seed** and see if this leads to a disorganization of the CMTs in the abaxial epidermis.
- We can **use the indenter to apply pressure on the seed for few hours** and see if this affects CMT organization in the abaxial epidermis. Note that, if it works, we will not have a large number of replicates with this experiment because we can only press one seed at a time
- We can perform **cell ablation in seeds at 2DPA** like we did for seeds at 5DPA and see if this leads to a circumferential reorganization of the CMTs.
- We can **compress isolated seeds between two coverslips** so that we can see the deformation that we apply to the seed and image the response of the CMTs.

According to the literature (such as Hamant *et al*, 2008 or Sampathkumar et al, 2014), we should start to see an effect of these perturbations on CMT organization after 4h. We thus propose to image **CMTs after 5h** to ensure

that we have a robust response. Note that we are proposing many different experiments because the seeds at 2DPA (during the anisotropic growth phase) are quite fragile so we do not know if we will be able to successfully carry all of these experiments.

Although they do not allow to see the direct response of CMTs to the application of mechanical forces, we also propose to do the following experiments to further support that mechanical forces control CMT organization in the abaxial epidermis:

- We will **compare CMT organization in the WT and in the *iku2* mutant at 1DPA**. This is motivated by our previous study supports that this mutant has increased endosperm pressure leading to more tension in the testa (Creff, Ali et al, 2023). We expect the CMTs in the abaxial epidermis to be more organized in *iku2* as we already showed in Fig. 5E that the seeds of the *iku2* mutant are elongating more than the WT between 0 and 2 DPA (which fits with our hypothesis).
- As previously described, we can **treat isolated seeds with isoxaben to inhibit cellulose deposition**, which has been shown to increase tension and lead to more organized microtubules. This response is however not fast. According to the literature, it should take at least 10 hours for the IXB to have a sufficient effect on the CW that the CMTs can reorganize (Heisler et al, 2010 and Sampathkumar et al, 2014),
- **Additional experiment:**

This was not asked by the reviewers but we also want to **show quantifications of the organization of the CMTs in the outer face of the abaxial epidermis from 0 to 3 DPA** (and not just 2 DPA like we already have) to study the dynamics of CMT disorganization and correlate it with the decrease in seed growth anisotropy that we see in Fig. 5E. This should really strengthen our claim that the transition from anisotropic to isotropic growth is due to a reduction in CMT organization in the abaxial epidermis.

Dear Dr. Landrein,

Thank you for providing a preliminary revision plan for your manuscript. I have now discussed it with reviewer #3, who found that the proposed experiments would address their concerns. Therefore, I would like to invite you to submit a revised manuscript in response to the reviewers' comments along the lines indicated in your revision plan. Please note that acceptance here will ultimately depend on the input of the referees on the revised version.

We generally allow three months as standard revision time, but this can be extended to six months for more extensive revisions. Should you foresee a problem in meeting this three-month deadline, please let us know in advance to arrange an extension.

As a matter of policy, competing manuscripts published during this period will not negatively impact on our assessment of the conceptual advance presented by your study. However, please contact me as soon as possible upon publication of any related work to discuss the appropriate course of action.

When preparing your letter of response to the referees' comments, please bear in mind that this will form part of the Review Process File and will therefore be available online to the community. For more details on our Transparent Editorial Process, please visit our website: <https://www.embopress.org/page/journal/14602075/authorguide#transparentprocess>. Please also see the attached instructions for further guidelines on preparation of the revised manuscript.

Please let me know if any further questions regarding the revision arise, and I look forward to receiving the revised manuscript!

With best regards,

Ieva

Ieva Gailite, PhD
Senior Scientific Editor
The EMBO Journal
Meyerohofstrasse 1
D-69117 Heidelberg
Tel: +4962218891309
i.gailite@embojournal.org

We realize that it is difficult to revise to a specific deadline. In the interest of protecting the conceptual advance provided by the work, we recommend a revision within 3 months (23rd Jan 2024). Please discuss the revision progress ahead of this time with the editor if you require more time to complete the revisions.

Response to the reviewers

Referee #1:

A very interesting, well written, and comprehensive study on the biomechanics of seed development. The way the results are presented, starting from the morphological features of growing seeds, to cellular growth differentials to a subcellular picture based on microtubule-reorientations, is appealing and I have enjoyed reading this manuscript. Even though the role of mechanics in morphogenetic processes in plants has been explored in a variety of organs, this study focusing on seed development is novel and timely. Moreover, the experiments are carefully described and conducted and support the conclusions of the paper. As such, this paper meet all requirements per the EMBO journal guidelines.

We thank the reviewer for supporting our manuscript and appreciating our work.

I have some small remarks/questions but these should be considered as minor:

- The authors start their story with a morphological characterization of seeds during their growth, revealing distinct stages. However, this is all based on a 2-dimensional measurement of an obviously three-dimensional object. Why was the third dimension not considered? Is it possible that taking this into consideration would have revealed additional stages, or is there some reason (un-beknownst to this reviewer) why the chosen two-dimensional projection is the only relevant one?

A1: We focused on the length and width of the seed because our size measurements were done in 2D. However, following this interesting comment from reviewer 1, we also looked at seed thickness and confirmed that the seed appears as a flattened ovoid whose thickness is lower than its width (Fig. EV1DE). Such differences between width and thickness are interesting, because they imply difference in stress anisotropy between the flatter sides of the seed and the flanks, according to the mechanical model we are now showing (Fig. 4C), which we were able to correlate with differences in the degree of anisotropy and the orientation of the CMTs between these two areas (Fig. 4F). However, we do not know whether the maintenance of the flatness of the seed also depends on CMT responses to forces. Zhao *et al*, 2020, Current Biology showed that a specific pattern of CMT organization in the transverse walls of the inner layers of the sepals contribute to the flattening of this organ. However, the case of the seed should be different because the inside is mostly filled with the large vacuole of the endosperm at the stages we consider. We looked at the transverse walls in the outer-integument and observed that the CMTs were always oriented perpendicularly to the seed surface in both abaxial and adaxial epidermis at 2 and at 5 DPA (Fig. 3A, 5A, 6A), which correlates with the reduced growth in thickness of these layers compared to the growth in length, but we do not know if this also affects overall seed thickness (Fig. EV2C).

- In their second section "Seed shape depends on..." Of the results and discussion, the authors refer to previous work in stating that seed size is the product of a mechanical balance between endosperm and seed coat. It made me wonder what the role of the embryo inside the seed is, which grows and develops in tandem; would a growing embryo not also provide an additional internal pressure in the seed whose action must be balanced at the exterior to reach a stable mechanical object?

A2: In the WT, the growth of the seed, which mainly occurs within the first week following fertilization, normally precedes that of the embryo, which starts at around 5 days post-fertilization (before this point the embryo is patterned but does not grow significantly). Although previous work has shown that the inner pressure of the seed decreases over time (Beauzamy et al, 2015; Creff et al,

2023), we know that the combined pressure of the endosperm and the embryo can put the seed under tension at later stages of seed development because a seed coat mechanosensitive gene (*ELA1*) was found in a transcriptomic analysis of *zhoupi*, a mutant in which the endosperm is not eliminated during embryo growth, thus generating mechanical conflict between the two internal seed compartments that together push on the seed coat (Creff et al, 2015). We also know that the seeds can continue to grow when the embryo is growing at the expense of the endosperm in seeds developing *in vitro* in a medium with high osmolarity, which delays the mechanosensitive stiffening of the seed coat (Creff et al, 2023). However, these phenomena occur after the time-window covered by our current manuscript, we thus do not expect the embryo to play a major role in promoting growth at this stage of development.

- In the same section, the authors remark that their data suggests that seed shape is dictated by the seed coat mechanics. Given that shape (geometry) in growing solid objects is usually the product of two (or more) opposing mechanical forces so that these can be balanced to produce a stable final state, this comment seems somewhat counter-intuitive to me, and may require elaboration.

A3: We agree with the reviewers and have corrected the text accordingly. Seed growth relies on a mechanical balance between the internal pressure of the expanding endosperm, that promotes growth, and the mechanical properties of the seed coat, that restrict it. In the section that the reviewer refers to, we meant that it is the seed coat that dictates the shape, as pressure is isotropic by nature.

- Perhaps nit-picking (given the wealth of literature on this topic in other tissues) but the authors experimentally establish a correlation between CMT alignment and the direction of anisotropic growth (The section starting with "We used the FibrilTool plugin"). Then they conclude a causality from this correlation in that CMT orientation determine the axis of seed elongation. Strictly, their data does not show that, it shows that the two are correlated, but this does not rule out that the anisotropic growth orients the CMTs, and not the other way around. Also, in the subsequent paragraph a similar observation can be made about correlation vs causation.

A4: We agree with the reviewers that we are mostly looking at CMT organization and not cellulose deposition, which is the process that confers anisotropic material properties to the wall. For this revision, we attempted to stain cellulose fibers in seed coat walls using the Scarlet 4B dye to confirm that their orientation correlates with that of the CMTs and is indeed perpendicular to the main seed axis (Anderson et al, 2010, Plant Physiology). However, fiber signals in the seed were not clear enough to allow us to draw any conclusion. Nevertheless, our interpretation that it is the CMTs that define the axis of seed elongation through the guided deposition of cellulose fibers is not only supported by the literature but also by the imaging and analysis of the seed shape defects of mutants involved in CMT organization (*ktn1* and *spr2*), in cellulose guidance by the CMTs (*csi1-3* and *ttl1 ttl3*) and in cellulose deposition (*prc1-1*) (Fig. EV3AB).

- In the description of the study of MT mutants (*ktn1*, *spr2*, etc.), the authors provide the names of the mutated genes, and then state that the observed effects are in line with the function of these genes, without making it clear to the reader what these factors are. I think, given that this is an important part of this study, this would be courteous. For example, explain that katanin is a MT severing factor so that a loss-of-function leads to a much less dynamic CMT array which also loses its sensitivity to mechanical stress, and then explain how this links to the observations in this study. The way it is currently written asks the reader (if they are not CMT experts) to go through a lot of literature to interpret some central findings in this paper.

A5: We apologize to reviewer 1 for not describing the mutants in enough details. As they were already described in many studies, we went too quickly over their description. This has been corrected in this new version of the manuscript. We also agree with the reviewer that, in the case of *ktn1* and *spr2*, it is not really clear how the disruption of these MAPs can lead to decreased sensitivity to mechanical stresses, even-though this has been clearly shown for other plant organs (Uyttewaal et al, 2012; Sampathkumr et al, 2014; Hervieux et al, 2016 and 2017; Wightman et al, 2013; Lin et al 2013 together with Tan et al, 2022). Understanding this would require modelling of the self-organization properties of the CMTs in the cell, and of how this is influenced by cell geometry, by MAPs such as *ktn1* and *spr2*, and by extrinsic signals such as mechanical forces.

Nevertheless, as we had lines expressing a CMT reporter in the *ktn1* mutant, we collected more data for CMT organization and response to forces in *ktn1* mutant seeds. We observed, as observed previously (ex: Uyttewaal et al, 2012 and Sampathkumr et al, 2014), that the CMTs facing the outermost cell wall of the seed coat of the *ktn1* mutant were less organized than in the WT and that their organization was, contrary to the case in the WT, not increased following compression (Fig.EV4F). Interestingly, we also observed that CMTs were less prone to orienting along the maximum axis of curvature in the *ktn1* mutant than they are in the WT (although this should be interpreted with the caveat that stress should be less anisotropic in *katanin* mutants as the seeds are rounder than WT seeds, Fig. EV3A-B). These observations fit well with our new data (Fig. 4A-B and Fig. EV4A-B) supporting that the effect of mechanical forces on CMT organization is dual: they increase the degree of organization per cell (i.e. the tendency of CMT bundles to orient along the same axis), and they increase the probability that CMTs orient along the maximum axis of curvature (See point **A9**).

- This paper is good and publishable as is. However, it did make me wonder, similarly as with other papers in the same spirit for other tissue types, if the connections between mechanics>CMTs>morphogenesis could not be study in a more causal way in systems in which CMTs can be disrupted in an inducible manner (e.g. using opto-genetics or chemical induction), so that one could directly manipulate the CMT at a particular phase in such a complex process, and see if the resulting effect is consistent with the hypothesis that is formulated. For example the opto-Katanin method for mammalian cells, would be a fantastic tool, but of course cannot be implemented in plants without re-engineering. I am not suggesting this should be done here, but I think it would be a valuable tool to get a significant step closer to a real proof of the causal relationships that are proposed in this paper, and others.

A6: We thank the reviewer for this interesting suggestion. We would love to do such experiment but unfortunately, optogenetics is more complex in plants than in animals (as the plants have to be in the light to grow). Being able to manipulate CMT dynamics in specific areas of the cell would be really useful to better understand how CMTs can be differently oriented, and react differently to specific cues, in different cells, or even in different faces of the same cell. Although it was not intended to be used for this purpose, the *pPDF1::mCitrine-MBD* line is a tissue-specific manipulation of CMTs organization in the seed coat layers as this line, that expresses a microtubule reporter in the abaxial epidermis of the outer-integument only, shows decreased CMT organization (at least compared to other CMT reporters), which correlates with the formation of rounder seeds than the control, fitting with our model that the abaxial epidermis is important for growth control (Appendix Fig. S2).

Referee #2:

In the paper entitled "Spatiotemporally distinct responses to mechanical forces shape the developing seed of Arabidopsis" Bauer et al. seek to address the challenging question of how organ-scale control

of anisotropy of seeds is controlled. They conduct a nice phenotypic analysis of the shape changes that occur at the organ scale. This is accompanied by cellular scale quantitative analyses of the growth behaviors that occur as seed growth behaviors transition from anisotropic to isotropic. This part of the paper is solid. The remainder of the paper that attempts to conclude that specific arrays of microtubules on specific faces of the seed epidermal cells sense mechanical stress to orchestrate the morphogenesis process. Overall, their conclusions are not supported by the data or by the cited prior publications that are used to justify the overall approach. Major weaknesses are explained below.

We are grateful that reviewer 2 finds our analysis of cell and organ growth in the seed interesting but regret that he was not convinced about the second part of the manuscript concerning the role of CMT response to force in the control of anisotropic growth. As we will describe in the following section, we have now performed several new experiments and provided a model to better support the hypothesis that CMT response to shape-driven stresses controls seed growth anisotropy.

The literature based in the field of mechanosensing is not covered in an accurate or complete manner. This pertains to the field in general and the specific functions of KATANIN and SPIRAL2 in this context. These errors lead to a failure to define the true knowledge gaps, a failure to justify their approach, and widespread over-interpretation of the data in this paper. These weaknesses in scholarship confuse the reader, hinders advancement of the field, and leads to reduced credibility. Several examples are provided below with the goal of helping the authors improve the paper.

We apologize that we were unable to cover the entirety of the literature in this manuscript. However, we do not agree with reviewer 2 that the literature that was not necessarily cited invalidates our model. We would respectfully point out that the literature mentioned by reviewer 2 mainly concerns pavement cells. These are a mechanically very different system from the seeds, notably because they are predominantly influenced by local or even autonomous cell-driven stresses, while the cells of the seed coat are also put under tension as a result of endosperm expansion, which leads to the development of organ level shape-driven stresses (Beauzamy *et al*, 2014, Creff *et al*, 2015 and 2023).

Scholarship issues:

1) The authors state:

"Mechanical forces have notably been shown to affect CMT organization, and thus cellulose deposition (Hamant *et al*, 2008; Sampathkumar *et al*, 2014; Robinson & Kuhlemeier, 2018; Hejnowicz *et al*, 2000), so that shape-driven stresses can affect anisotropic growth in a variety of plant organs (Trinh *et al*, 2021)."

These papers in isolation or in aggregate do not make clear connections between stress, microtubules, cellulose, and growth patterns. The authors fail to cite more recent papers that make mechanistic contributions in this area.

A7: We apologize if this was not clear in the way that the manuscript was written but the four first references cited: "Hamant *et al*, 2008; Sampathkumar *et al*, 2014; Robinson & Kuhlemeier, 2018; Hejnowicz *et al*, 2000" indeed do not all show clear connections between stress, microtubules, cellulose and growth patterns. They do, however, all give support to the fact that CMTs can respond to forces in different plant organs. However, the references cited in the review Trinh *et al*, 2021 do make connections between stress, CMT, cellulose and growth patterns. We now directly cite these key articles (Hervieux *et al*, 2016 and 2017, showing that CMT responses to forces are involved in the

control of the shape and shape variability in sepals; Uyttewaal et, 2012, showing that CMT responses to forces are involved in the control of SAM shape (and notably the correct folding of the boundary), and Zhao et al, 2020, showing that CMT responses to forces are involved in the shaping of the leaf blade). Although readers are free to question the results and interpretation of all of these studies, they try, like our study, to make the link between CMT responses to forces and cell and organ growth.

2) The authors state:

"In many aerial organs, it is believed that growth is promoted by the pressure of inner tissues but restricted by the mechanical properties of one or several outer layers, which is often the epidermis (Kutschera & Niklas, 2007). "

This ignores turgor-generated forces in the epidermis that contribute to expansion of the tissue and the organ. The authors fail to account for recent papers that analyze in detail the relationships between stress, microtubules, and cell wall anisotropy in the epidermis that can strongly influence the growth of the tissue.

Zhao, F., Du, F., Oliveri, H., Zhou, L., Ali, O., Chen, W., Feng, S., Wang, Q., Lu, S., Long, M., et al. (2020). Microtubule-Mediated Wall Anisotropy Contributes to Leaf Blade Flattening. *Curr Biol* 30, 3972-3985 e3976.

Belteton, S., Li, W., Hatam, F.A., Quinn, M.I., Szymanski, M.R., Marley, M., Turner, J.A., and Szymanski, D.B. (2021). Real-time conversion of tissue-scale mechanical forces into an interdigitated growth pattern. *Nature Plants* 7, 826-841.

Schneider, R., Ehrhardt, D.W., Meyerowitz, E.M., and Sampathkumar, A. (2022). Tethering of cellulose synthase to microtubules dampens mechano-induced cytoskeletal organization in *Arabidopsis* pavement cells. *Nat Plants* 8, 1064-1073.

A8: We do not agree with the interpretation of reviewer 2, but would like to express out thanks for enabling us to pinpoint areas that may not have been clear. All turgid cells are indeed submitted to pressure-driven stresses and, in the case of the pavement cell, this stress combined with the complex lobed shape of the cell, generates a local pattern of stress to which CMTs seem to respond (according to Sampathkumar et al, 2014 and indeed supported by the work of Belteton et al 2021, and Schneider et al, 2022). In the case of the pavement cell, it is thus local stresses that seem to preferentially orient CMTs, and the patterns the CMTs exhibit is subcellular and dependent of the shape of the cell (which is very complex compared to that of the cells in the seed coat) rather than supracellular. In the case of the seed, the cells of the seed coat are not only put under tension by their own pressure but also by the pressure of the expanding endosperm, thus leading to the generation of additional stresses that are dependent on the overall shape of the seed . As we now describe more clearly in the manuscript, the fact that the seed coat is placed under tension by endosperm expansion is experimentally supported by the work of Beauzamy et al, 2014 showing that most of the rigidity of the seed at early stages of development comes from endosperm pressure (that can be released by pricking the endosperm, as visualized by our new Appendix Movie S1). It is also supported by our recent study (Creff *et al*, 2023) showing that increased endosperm pressure in the *iku2* mutant leads to more tension in the seed coat, which is sensed by the adaxial epidermis of the outer-integument and leads to an early stiffening of seed coat walls and a precocious restriction of growth. This claim is now also further supported by another new experiment performed for this revision showing that cells in the seed coat abaxial epidermis behave like "rubber bands" when endosperm pressure is released by pricking the seed: they decrease in length and width but increase

in thickness, because the endosperm pressure that normally puts them under tension has been released (Fig. EV4D).

Regarding the paper of and Zhao and colleagues (2020), this very interesting work indeed shows that CMT responses to forces in internal layers is involved in the control of the growth of the leaf (and notably its flattening). This work is not in disagreement with ours as the proposed model still shows that it is shape-driven stresses that generate the supracellular pattern of anisotropic growth seen in early developing leaves, by orienting the CMTs. It is however interesting to note that the case of the seed differs from the case of the leaf, as at early stages, there is only one giant, polynucleate cell with a large central vacuole (the endosperm) inside the seed, instead of the lattice of cells that can respond dynamically to forces that is found in the leaf primordium. Still, we agree that transverse walls could also be important for seed growth. We have thus now studied the orientation of the CMTs in transverse walls in the outer seed-coat cell layer and showed that they are always oriented perpendicularly to the seed surface (Fig. 3E, Fig. 5A, Fig. 6A), which we correlate with the fact that the growth in thickness of the layers is reduced, at least compared to the growth in length (Fig. EV2C-D). In the abaxial epidermis during the anisotropic growth phase only, we also measured that there were more CMT bundles in transverse walls that were parallel to the main seed axis than in transverse walls that were perpendicular to the main seed axis (Fig. EV3C). These CMTs could thus, together with the those of the longitudinal walls that are also preferentially oriented perpendicularly to the main seed axis, and like hoops around a barrel, restrict growth perpendicularly to the main seed axis during the anisotropic growth phase (Fig. EV3D).

3) The authors state:

"In pavement cells, differences in mechanical properties between contiguous walls, also associated with local pectin accumulation, trigger the initiation of the lobes through buckling, which are then amplified through CMT responses to forces (Majda et al, 2017; Sampathkumar et al, 2014). " Although not directly related to the topic of organ-scale morphogenesis these statements inaccurately portray the field of stress sensing and morphogenesis. The papers cited below do not prove the points the authors are making. The role of pectin in lobe formation is not supported by clear experimental data and should not be used as an illustrative example.

A8: The reference to the manuscript of Sampathkumar et al, 2014 was included to support that the shape of leaf cells depends on CMT responses to forces. However, the work of Majda et al, 2017 presents modelling work to support those heterogeneities in mechanical properties along and across anticlinal walls could initiate the interdigitated pattern of leaf cells. The authors also experimentally observed these heterogeneities by atomic force microscopy and correlate them with local changes in pectin composition. Nevertheless, we have tempered our phrasing to state that this model still needs additional experimental support.

4) In the results section, the authors state:

"Given that this latter class of seeds have phenotypically wildtype endosperm and embryos (being heterozygous), but a phenotypically mutant seed coat owing to its maternal origin (Fig 1E), our data suggest that seed shape mainly depend on the mechanical properties of the seed coat. " The involvement of KTN1 and SPR2 in stress sensing is vastly overstated and this leads to an overinterpretation of the mutant phenotypes in this paper. This is a major problem. The phenotypic data do not say anything direct about the mechanical properties of the seed coat. It just means maternal tissues require wild type KTN and SPR2. The connections between ktn and spr2 phenotypes and stress sensing are not clearly proven in this paper (see below).

A8: The point we wanted to make through the reciprocal crosses was that it is the seed coat (and not the endosperm or embryo) that determines the shape of the seed. This is a valid conclusion since (as it is actually the case for all the recessive seed-shape mutants we analyzed) mutants such as *ktn1* and *spr2*, even when they are crossed with WT pollen so that the endosperm and the embryo are heterozygous, maintain their seed shape defects. This is logical as, in our growth model, the coenocytic endosperm promotes growth through pressure, which is isotropic by definition, and since, as discussed for reviewer 1, the embryo does not impact early seed growth. The statement that the link between stress sensing and KATANIN and SPIRAL is “overstated” like reviewer 2 claims is however questionable, as it will be discussed in point **A9**.

Technical issues:

1) The paper claims to link stress, microtubule patterns, and anisotropic growth. In the experimental system the stress patterns are not known, neither is the critical cell face of the cell known. The authors state:

"CMTs determine the main axis of cell elongation by controlling the oriented deposition of cellulose microfibrils in the wall (Paredes et al, 2006). In leaves, sepals and meristems, the CMTs facing the outermost cell wall of the epidermis, which is believed to be load-bearing, organize according to shape-driven stresses and are believed to control the anisotropic growth of these organs (Hamant et al, 2008; Hervieux et al, 2016; Sampathkumar et al, 2014; Zhao et al, 2020). However, in hypocotyls, it is the CMTs that face the inner side of the epidermis that are robustly oriented perpendicularly to the main axis of elongation (Crowell et al, 2011; Chan et al, 2011). This suggests that the inner wall of the epidermis is load-bearing and controls the elongation of this organ. This scenario is also consistent with computational simulations and experiments on hypocotyls (Robinson & Kuhlemeier, 2018; Verger et al, 2018) (Robinson 2018, Verger 2018)."

The authors rely on a cellular scale anisotropy measurements and an assumed correlation with stress to conclude which cell face mediates stress dependent anisotropic growth. First, it may not be stress dependent control. There are many mechanisms by which organ scale shape change can occur. The authors rely on a superficial correlation between microtubule order and an assumed stress pattern based on cell curvature to make their conclusions. If the process is completely microtubule-mediated (not shown here) the authors really don't know what microtubule array is responsible. They ignore the possible contribution of anisotropy in the anticlinal walls, which is known to affect anisotropic growth in leaves. In addition, the authors have no clear idea of the stress patterns in the cells/tissue. They assume that stress is related to local curvature, but many other factors besides local curvature could define the stress patterns in the cell. Curvature just correlates with the microtubule pattern. It is also unclear what the stress patterns of the seed will be upon squeezing. There are likely combinations of compression and tensile forces that change upon squeezing and this is impossible to intuit. Some kind of reliable FE model would be needed to even attempt a guess.

A9:

As reviewer 2 point out, we indeed use a correlative approach to deduce which layer of the outer-integument controls the growth the seed (as the *ap2* mutant seed is round and AP2 is known to control outer-integument cell identity). We reasoned that if the CMTs control seed growth anisotropy through the guidance of cellulose fiber deposition (which is supported by the seed shape phenotype of *ktn1*, *spr2*, *csi1-3*, *ttl1ttl3* and *prc1-1*), then we should find CMTs that orient (and could guide cellulose deposition) perpendicularly to the main axis of seed elongation in at least one layer of

the outer-integument (given the *ap2* phenotype). We found that only the cells of the outer cell layer had CMTs organized and preferentially oriented perpendicularly to the main axis of seed elongation supporting that they should be those controlling anisotropic seed growth. In the first manuscript, this analysis was only done in longitudinal walls (facing the inner or outer side of the seed in each layer). However, in light of this comment, we also now analyzed the transverse walls of the first layer (Fig. 3E, Fig. EV3C). The result of this analysis, already described in point **A8** supports that the transverse walls of the first layer could participate, together with the longitudinal walls, in the control of seed elongation during the anisotropic growth phase (Fig. EV3D).

Regarding the claim that these CMTs orient according to shape-driven stresses in the first layer, in addition to the experiments described in point **A8** and further supporting that the seed coat is indeed put under tension by endosperm expansion, we also performed simulations to analyze the correlation between curvature and stress direction on an inflated shell whose shape is similar to one of WT seeds at 2 DPA (obtained by measuring seed length, width and thickness and fitting ellipses in different directions) (Appendix Supplementary Methods: Model). These simulations show, as expected, that there is a strong correlation between the orientation of curvature and the main stress direction in the seed, and that in almost all the areas of the seed (except the tip), the main direction of stress is perpendicular to the main seed axis (Fig. 4C). Interestingly, the model also predicts that the stress is more anisotropic in the flank of the seed (an area that is more curved) than in the flatter sides of the seed (Fig. 4F). Strikingly, we show that the CMTs in the flank of the seed are indeed more organized and more preferentially oriented perpendicularly to the main seed axis and thus to the main stress direction predicted by our model (Fig. 4F). Together, these findings reinforce the claim that curvature can indeed be used to assess shape-driven stresses and that it strongly correlates with CMT orientation in the first layer.

Regarding the response to stress itself, we agree with the reviewer that it is difficult to precisely define what is going on when we compress seeds within the fruit for 24h, and thus to know how forces affect the CMTs in this context (other than increasing the degree of CMT organization per cell). This is why we performed more direct and rapid applications of mechanical forces to the seeds. We performed ablation experiments where we killed a few cells of the first layer with a needle and observed, similarly to several previous studies (such as Hamant et al, 2006, Sampathkumar et al, 2014 or Malivert et al, 2021) a circumferential reorientation of the CMTs after only 5h (Fig. 4A). We also performed 20 μ m deep indentation of the seed with an indenter for only 5 to 10 minutes and again observed a circumferential reorientation of the CMTs around the indented area of the seed (top of the seed in confocal stacks) after 5-7 hours (which is the expected behavior according to Louveaux et al, 2016, Plant Journal) (Fig. 4B and Appendix Supplementary Fig. S3). Finally, we also observed that when we put the seeds in liquid media, they quickly inflate so that cells in first layer increase their size by around 25-40% in only 5h. Interestingly, this also leads to increased CMT organization per cell and an increased tendency for CMTs to orient perpendicularly to the main seed axis (and thus to the main stress orientation according to our model) (Fig. EV4B). Taken together, these experiments and the measurements of CMT organization in different areas of the seed, support the idea that mechanical forces promote the organization of the CMT in the cells and their preferential orientation along the main stress direction.

2) The *ktn* and *sprl2* mutant analyses cannot be linked to stress sensing and microtubule reorganization because there are no strong published data indicating that either of these genes function in that capacity. This point is alluded to above. In the absence of such data, either explained clearly based on a published result or new data, the interpretation of the mutant phenotypes is flawed.

A10: The claim that there is no strong data published indicating that *KATANIN* and *SPIRAL2* are involved in CMT response to stress seems rather surprising given the following papers published in the last 12 years:

- Uyttewaal et al, 2012, Cell. Mechanical stress acts via KATANIN to amplify differences in growth rate between adjacent cells in Arabidopsis.
- Eng et al, 2021, Current Biology. KATANIN and CLASP function at different spatial scales to mediate microtubule response to mechanical stress in Arabidopsis cotyledons.
- Hervieux et al, 2017, Current Biology. Mechanical Shielding of Rapidly Growing Cells Buffers Growth Heterogeneity and Contributes to Organ Shape Reproducibility.
- Hervieux et al, 2016, Current Biology. A Mechanical Feedback Restricts Sepal Growth and Shape in Arabidopsis
- Wightman et al, 2013, Current Biology. SPIRAL2 determines plant microtubule organization by modulating microtubule severing
- Lin et al, 2013, Current Biology, Rho GTPase signaling activates microtubule severing to promote microtubule ordering in Arabidopsis TOGETHER WITH Tang et al, 2022, Current Biology, Mechano-transduction via the pectin-FERONIA complex activates ROP6 GTPase signaling in Arabidopsis pavement cell morphogenesis
- Sampathkumar et al, 2014, eLife, Subcellular and supracellular mechanical stress prescribes cytoskeleton behavior in Arabidopsis cotyledon pavement cells
- Zhao et al, 2020, Current Biology, Microtubule-Mediated Wall Anisotropy Contributes to Leaf Blade Flattening
- Louveaux et al, 2016, The Plant Journal, The impact of mechanical compression on cortical microtubules in Arabidopsis: a quantitative pipeline

If none of the manuscripts mentioned above contain data that convince the reviewer that either SPR2 or KATANIN, or both, are involved in CMT responses to forces, providing convincing data would appear to be a difficult undertaking. Nevertheless, as described in point **A5**, as we had introgressed the CMT reporter into the *ktn1* mutant, we collected more data on CMT organization and responses to forces in the seeds of this mutant. We observed, as has been observed in other organs (ex: Uyttewaal et al, 2012 and Sampathkumar et al, 2014), that the CMTs facing the outermost cell wall of the seed coat of the *ktn1* mutant were less organized than in the WT and that their organization was, contrary to that in WT, not increased following compression. Interestingly, we also observed that they CMTs were less prone to orient along the maximum axis of curvature in the *ktn1* mutant than they are in WT seeds (again with the caveat that this could be because stress anisotropy is lower in *ktn1* as the seeds are rounder than WT seeds) (Fig. EV4F).

Referee #3:

In this study, Bauer et al investigate the development of seeds of Arabidopsis in context of growth trajectories and mechanobiology. The authors use growth series of Arabidopsis seeds to conclude that growth follows an early anisotropic phase and then switches to an isotropic phase. They show that the first phase is accompanied and largely dependent on microtubule organization in the outer cell layers of the seed coat and that the second growth phase is largely dependent on inner cell layers of the seed coat and the microtubule organization in these layers. The work is largely well conducted and logically written. While I do find the study compelling, I have some comments that I think is necessary for both this study and the field of mechanobiology and the corresponding microtubule organization analyses in general.

We thank reviewer 3 for this positive assessment of our work.

Major points:

While I applaud the authors for using multiple fluorescence microtubule marker lines to analyze and quantify microtubule organization, I am still left with a sense of uncertainty regarding the conclusions. This sense is actually something I have had in context to several other studies of microtubule quantification with the fibril tool and stress applications. In the different figures of microtubule fluorescence, I am unable to see any real differences. This is even in the blow-ups of images I obtained from the EMBO Journal for reviewing purposes. I am of course aware of the quantification that is associated with this, but I would feel a bit uncomfortable to support conclusions based on the images that are supplied. For example, Figure 3: I cannot say that the images here really support the claims as I cannot see really clear microtubules at least in the outer face and inner face of the adaxial epidermis. In addition, the microtubule alignments appear high in the abaxial outer face cells, but many of these are not aligned with the growth axis, and so this is a bit troubling. I still have to point out that I do not doubt the conclusions of the paper and I think the authors have done a better job than many other studies in this field. I am just saying that with these types of images, I am not confident about quantification and how well a fibril tool does. I will leave it up to the authors to argue for the use of these or not. In this context, I would like the authors to zoom in and convince the readers that the microtubules can clearly be seen in cells that they use for quantifications.

A11: We apologize to reviewer 3 that the figures provided in the first submission got compressed so that we could not see the microtubules well. We now provide pictures of higher resolution and closeups on different cells so that the organization of the CMTs and their correlation with the orientation given by the Fibriltool software can be better visualized. Most of the stacks where we visualized CMTs were acquired on a high-resolution microscope (Zeiss 980 equipped with the Airyscan technology). However, some of the initial stacks were also acquired on other microscopes that are not high resolution and where the CMT signal is lower (Leica SP8 and Nikon C2). Although Fibriltool is not supposed to be dependent on the quality of the picture (Boudaoud et al, 2014, Nature Protocols), we checked whether the degree of CMT organization given by the plugin was dependent on the type of microscope that was used to acquire the data. We found that Fibriltool gave the same degree of organization of the CMTs for the same seeds that were sequentially imaged using the Zeiss 980 with Airyscan and using the SP8 (Appendix Supplementary Figure S1).

Regarding the fact that we see cells in the first layer whose CMTs are not oriented perpendicularly to the main seed axis, this is not only seen in the pictures but also in our quantification of the averaged CMT orientation per cell compared to the main seed axis, showing some variability of angle frequency but still with a peak that is close to 90° for the CMTs in the first layer. This is logical given that the zones of the seed that we used to study CMT organization in different layers and with different CMT reporters are the flatter sides of the growth zone, which, according to the new model we developed and as validated experimentally, are submitted to less anisotropic stresses than the flanks of the seed (Fig. 4F). As the anisotropy of the stresses seems to increase the degree of organization of the CMTs but also their probability of orienting along the maximum stress direction, it is logical to see some cells whose CMTs are not preferentially oriented perpendicularly to the main seed axis in single snapshots even though they are still, on average, preferentially oriented along this axis. This is also logical given the fact that we observed that the CMTs are very dynamic within the cell and could quickly reorganize when we carried out our comparisons between microscopes or when we followed the same seed over short periods of time.

Minor points:

The authors may want to exchange some images as for example the Figure 6, EV3 and EV6 contain the exact same images. This is not indicated and I would assume the authors have more than these images to convince the reader (please recheck the rest of your figures to avoid any such duplications). If published this could have some rather unwanted issues. 7

A12: We use the same pictures in these figures on purpose to show that it is the same data that are used in these different figures (the previous Fig.EV3 shows the comparison of the different CMTs markers at 2DPA, while Fig.6 shows the comparison of the different markers at 2 vs 5 DPA, and Fig.6 shows comparisons between 2 and 5 DPA of the MAP65-1-RFP reporter only). As this could trouble the readers in the same manner as it troubled reviewer 3, we know show different pictures, even if they are of the same marker and at the same time-point.

Some spelling mistakes in figures need to be corrected (for example Anisotropic in Fig 2).

Go through the manuscript and check some spelling etc...for example MAP61 instead of MAP65 etc.

A13: Thanks for pointing out these errors, that have been corrected.

The reduction of cellulose and MT organization virtually always result in loss of anisotropic growth, so not sure how much value this really is to the study?

A13: We agree with reviewer 3 but this could actually be questioned by some readers and we find quite interesting that all of the mutants involved in CMT organization and guidance of the cellulose deposition we looked at shows defect in anisotropic growth in the seed, which is not always the case for other plant organs. The *ttl1 ttl3* mutant, for instance, only shows defects in root growth under stress conditions (Kesten et al, 2022, Science Advances) while loss of *CS11* does not necessarily generate completely isotropic organs like it does in the seed (Bringmann et al, 2012, The Plant Cell).

The authors talk about isotropic microtubule array in cells facing the inner side of the seeds during the isotropic growth face, but then say that they are somewhat longitudinally organized. These statements are in contrast to each other.

A14: We apologize for this mistake: the CMTs facing the inner side of the second layer (wall 3) are essentially disorganized (low degree of CMT organization, at least compared to the outer face) but they tend to have a small preference to be oriented longitudinally. To what extent this preferential orientation matters given the low degree of organization (meaning that most of the cellulose fibers, if their deposition is still controlled by CMTs, will be preferentially oriented longitudinally) is questionable. We do see a small tendency for the seeds to grow slightly more in width than in length at later stages of development (Fig. 1D), but this could simply be a geometrical effect induced by the fact that inflating an organ with isotropic material properties would tend to make it become rounder, which is what we observe.

The authors talk about "slightly less" preferentially organized...what does this mean??

A14: This is a fair point, we replaced these statements with numbers (for instance putting the difference in mean orientation, etc.).

Dear Benoit,

Thank you for submitting the revised version of your manuscript to The EMBO Journal. Your study has now been seen by one of the original referees, who finds that their previous concerns have been addressed and now recommends acceptance of the manuscript. There now remain only a few editorial points that need addressing before I can extend formal acceptance of the manuscript:

1. Please check that the funding information is correct and identical both in the manuscript and our online system. Currently the funding for PhD theses mentioned in the manuscript is not included in our online system.
2. Please submit a complete author checklist, which you can download from our author guidelines (<https://www.embopress.org/pb-assets/embo-site/EMBO%20Press%20Author%20Checklist-1642513524327.xlsx>). Please insert information in the checklist that is also reflected in the manuscript. The completed author checklist will also be part of the Review Process File.
3. CRedit has replaced the traditional author contributions section because it offers a systematic, machine-readable author contributions format that allows for more effective research assessment. Please remove the Author Contributions from the manuscript and use the free text boxes beneath each contributing author's name in our online submission system to add specific details on the author's contribution. More information is available in our guide to authors.
4. Please add a "Disclosure and competing interests statement" section before "References" (further info: <https://www.embopress.org/page/journal/14602075/authorguide#conflictsofinterest>).
5. We can accommodate up to five Expanded View (EV) figures. Please consider turning one of the EV figures into a main figure or adding it to the Appendix.
6. Please rename the movie file into "Movie EV1" and zip it together with a readme file containing its legend. Please remove the movie legend from the Appendix file.
7. Figure panels 2B and 6B-C are not mentioned in the manuscript text, please add the corresponding callouts.
8. Please remove the duplicate "Data and code availability" section from the Appendix file.
9. Our data editors have flagged the following issues in figure legends that need correcting:
 - Please note that the figure EV 6f does not contain any micrograph, perhaps the scale bar information refers to figure EV6g?
 - Please define the annotated p values ****/****/**/* in the legend of figure 1e; 5b; 6b; EV 2d; EV 3b, e; EV 4a-b, d, f; EV 5b-c; EV 6b, f; as appropriate.
 - Please indicate the statistical test used for data analysis in the legend of figure EV 1e.
 - Please define the box plots in terms of minima, maxima, centre, bounds of box and whiskers, and percentile in the legends of figures 1c-f; 3b; 4f; 5b; 6b, d; EV1a-b, e; EV2a-b, d; EV3b-c, e; EV4a-b, d, f; EV5b-c; EV6b, f.
 - Please add the number of replicates in the legend of figure EV1e.
 - Please define the error bars in the legends of figure 5c; EV1c.
 - Please define the orange arrows and red lines in the legend of figure 5a; 6a.
10. Papers published in The EMBO Journal are accompanied online by a 'Synopsis' to enhance discoverability of the manuscript. Please submit a short (1-2 sentences) summary of the findings and their significance in addition to the already provided bullet points highlighting the key results. Please also send us a synopsis image that is 550x300-600 pixels large (width x height, jpeg or png format). You can either show a model or key data in the synopsis image. Please note that the image size is rather small and that text needs to be readable at the final size.

With best wishes,

Ieva

We realize that it is difficult to revise to a specific deadline. In the interest of protecting the conceptual advance provided by the work, we recommend a revision within 3 months (23rd Jul 2024). Please discuss the revision progress ahead of this time with the editor if you require more time to complete the revisions.

Referee #3:

I have reviewed an earlier version of this manuscript and I think the authors have done a good job in addressing my concerns.

The authors addressed the remaining editorial issues.

Dear Benoit,

Thank you for addressing the remaining source data issues. I am now pleased to inform you that your manuscript has been accepted for publication. Congratulations on a nice study!

Before we forward your manuscript to our publishers, I would like to propose minor edits in the manuscript abstract and synopsis (please see below and the attached manuscript text file). I have also written a short blurb that will accompany the title of your manuscript in our online system. Please let me know if any corrections are needed.

Blurb:
Arabidopsis seeds are shaped by different responses of cortical microtubules to mechanical stress in their two outer layers.

Synopsis:
In plants, organ and tissue growth is regulated by mechanical force-dependent rearrangement of cortical microtubules and resulting changes in cell wall material deposition. This study shows that, in the model plant Arabidopsis, seed shape is the product of two distinct mechanical responses that are triggered in different layers of the seed coat outer layers at different stages of development.

- Seed growth proceeds via an initial stage of anisotropic growth, followed by isotropic growth later in development.
- Anisotropic seed growth depends on cortical microtubule response to shape-induced mechanical stresses in the outer layer of the seed coat.
- Isotropic phase of seed growth depends on cortical microtubule reorganisation in the inner layer of the seed coat.
- The switch to the isotropic growth mode is accompanied by reduced mechanosensitivity of cortical microtubules.

If you have any questions, please do not hesitate to contact the Editorial Office. Thank you for this contribution to The EMBO Journal and congratulations on a successful publication!

With best wishes,

Ieva
